# Mafa-dependent GABAergic activity promotes mouse neonatal apneas

Laure Lecoin [1,2,6 ✉], Bowen Dempsey[2,7], Alexandra Garancher [1], Steeve Bourane [3,4], Pierre-Louis Ruffault[5], Marie-Pierre Morin-Surun[2], Nathalie Rocques[1], Martyn Goulding[3], Alain Eychène [1], Celio Pouponnot [1✉], Gilles Fortin [2,7✉] & Jean Champagnat[2]

While apneas are associated with multiple pathological and fatal conditions, the underlying molecular mechanisms remain elusive. We report that a mutated form of the transcription factor Mafa (Mafa$^{4A}$) that prevents phosphorylation of the Mafa protein leads to an abnormally high incidence of breath holding apneas and death in newborn Mafa$^{4A/4A}$ mutant mice. This apneic breathing is phenocopied by restricting the mutation to central GABAergic inhibitory neurons and by activation of inhibitory Mafa neurons while reversed by inhibiting GABAergic transmission centrally. We find that Mafa activates the Gad2 promoter in vitro and that this activation is enhanced by the mutation that likely results in increased inhibitory drives onto target neurons. We also find that Mafa inhibitory neurons are absent from respiratory, sensory (primary and secondary) and pontine structures but are present in the vicinity of the hypoglossal motor nucleus including premotor neurons that innervate the geniohyoid muscle, to control upper airway patency. Altogether, our data reveal a role for Mafa phosphorylation in regulation of GABAergic drives and suggest a mechanism whereby reduced premotor drives to upper airway muscles may cause apneic breathing at birth.

[1] Institut Curie CNRS UMR 3347, Inserm U1021, UPSaclay, bat 110 Centre Universitaire, 91405 ORSAY, France. [2] UMR 9194 CNRS Neuro-PSI, Institut de Neurobiologie Alfred Fessard, 1 avenue de la terrasse, 91190 Gif-sur-Yvette, France. [3] Molecular Neurobiology Laboratory, The Salk Institute for Biological Studies, La Jolla, CA 92037, USA. [4] UMR DéTROI U1188 Université de La Réunion-CYROI 2, rue Maxime Rivière 97490 Sainte Clotilde La, Réunion, France. [5] Developmental Biology and Signal Transduction Group, Max Delbrück Center for Molecular Medicine, 13125 Berlin, Germany. [6] Present address: CIRB, CNRS UMR7241 Collège de France 11 place Marcellin Berthelot, 75005 Paris, France. [7] Present address: Institut de Biologie de l'École Normale Supérieure (IBENS), École Normale Supérieure, CNRS, INSERM, PSL Université Paris, 75005 Paris, France. ✉email: laure.lecoin@college-de-france.fr; celio.pouponnot@curie.fr; gilles.fortin@bio.ens.psl.eu

The neural control of breathing is affected in many pathological conditions such as Congenital Central Hypoventilation Syndrome[1], Rett syndrome[2,3], Multiple System Atrophy[4] and presumably in Sudden Infant Death Syndrome (SIDS)[1]. Although the etiologies are varied and depend on the age, apneic events are typical traits of these syndromes. Furthermore, the biggest public health challenges that involve apnea, beyond the increasingly recognized contribution of central and obstructive apneas to "sleep disordered breathing" or "complex apnea"[5], are neonatal apnea in the premature child[6]. While the pathophysiological consequences of apnea are fairly well described, the underlying regulatory molecular mechanisms remain elusive.

Active interruption of the respiratory airflow is essential in a number of neonatal behaviors, for example to direct milk toward the alimentary tract[7], to protect airways from the inhalation of dust or irritant chemicals[8], or to increase subglottal pressure before uttering a sound[9]. When their duration extends beyond the point at which blood oxyhemoglobin de-saturates, respiratory pauses can be life threatening. While central apneas refer to a failure of the central respiratory rhythm generator to maintain respiratory frequency, obstructive apneas, the most prevalent type of apneas, are characterized by a stop of the airflow despite persistent respiratory efforts, due to collapse of the upper airway (reviewed in Ref. [10]). Sleep-dependent reductions of pharyngeal muscles tonicity are known to increase the resistance of upper airways[11,12]. Pharyngeal constrictor muscles have only a marginal effect on upper airway closure[13] but the main pharyngeal dilator muscles - the genioglossus and geniohyoid muscles - that pull forward the hyoid bone are crucial for the maintenance of upper airway patency[14,15]. Benzodiazepines exacerbate oropharyngeal obstructions by depressing the motor drive in hypoglossal and recurrent laryngeal nerves that innervate upper airway muscles, incriminating overactive GABAergic neuronal drives in apneic episodes[16,17]. However, the endogenous mechanisms that may potentiate synaptic inhibition to cause apneic breathing remain unclear, as well as the inhibitory neuronal substrate.

Mafa is a member of the large MAF family of transcription factors (TFs) that also includes Mafb, c-Maf and Nrl[18,19]. MAF factors belong to the larger b-ZIP TFs of the AP1 superfamily (like c-Fos or c-Jun), but harbor a specific DNA binding domain that participates in the recognition of the Maf Responsive Element (MARE). MAF activity is tightly controlled post-translationally[20,21]. More precisely, phosphorylation is a key regulatory mechanism, as shown by the Ser64Phe human mutation that impairs Mafa phosphorylation while causing familial diabetes mellitus and insulinomatosis[22]. This mutation prevents Mafa phosphorylation on Ser65 by a priming kinase and the subsequent phosphorylation of Mafa by GSK3 kinase on 4 Ser/Thr residues. These residues (S61, T57, T53, S49) are highly conserved among MAF family members, and their phosphorylation mediates proteasome-dependent degradation of the protein through the proteasome[21,23]. While Mafa is mainly known to control insulin gene transcription in β-cells of the pancreas[23,24], it is also expressed in restricted neuronal populations of the peripheral nervous system and spinal cord[25–29] as well as in the brainstem (this study).

To understand the neurophysiological role of Mafa, we have generated conditional knock-in Mafa-4A mutant mice in which the 4 Ser/Thr residues are mutated into alanine. As a consequence, the mutant Mafa protein can no longer be phosphorylated by GSK3 nor be degraded by the proteasome, leading to intracellular accumulation and modification of its transactivation activity[21]. At birth, Mafa-4A mutants show an abnormally high incidences of apneas and died within 48 h. Moreover, we find that Gad2 transcription is a mechanistic target of Mafa

dysregulation and propose that increased inhibitory premotor drive to upper airway muscles may be causal to the respiratory deficit.

## Results

**Mafa phosphorylation is essential for postnatal survival**. To examine the physiological roles of Mafa, we have generated two cre-dependent Mafa knock-in mouse lines in which the Mafa coding sequence is either replaced by LacZ (Mafa^floxLacZ) or by a mutated Mafa4A allele (Mafa^flox4A, Supplementary Fig. 1a–c) encoding a non-phosphorylatable form of the Mafa protein. Both lines were crossed to the PGK^Cre line to obtain Mafa^LacZ/LacZ knock-out and Mafa^4A/4A knock-in mice, respectively. Mafa immunoreactivity in wildtype mice was lost in Mafa^LacZ/LacZ mice and, conversely, was increased in Mafa^4A/4A mutants, in keeping with the longer half-life of the non-phosphorylatable form of the Mafa-4A protein which can no longer be degraded by the proteasome (Supplementary Fig. 1d). As previously described by us and others[23,24,27], Mafa^LacZ/LacZ knock-out animals were viable, fertile and breathed normally. In contrast, no living Mafa^4A/4A mutants were recovered at weaning stage from intercrosses of Mafa^4A/+ mutants generated from two independent ES clones (Supplementary Table 1). At birth (postnatal day zero, P0), Mafa^4A/4A mutant pups displayed normal weight, general morphology and were able to suckle and vocalize (Supplementary Table 2). They were however hypoactive and apart from seldom jerks or twitches, did not show postural nor chest wall movements suggesting exacerbated inspiratory or expiratory efforts. They developed cyanosis and died within 12 h (Supplementary Fig. 1e, f). The viability of Mafa null pups in face of the rapid death of Mafa^4A/4A mutant pups suggested that lethality may be correlated to gene dosage and the overall Mafa protein levels. In support of this, Mafa^LacZ/+ carrying one Mafa^WT allele or Mafa^LacZ/4A pups with a single Mafa^4A mutant allele were fully viable, while heterozygous Mafa^4A/+ pups carrying one Mafa^4A allele and one Mafa^WT allele had a 60% survival rate at P21 (Supplementary Table 1).

Altogether, these data demonstrate that the Mafa4A allele is not itself lethal and that the abrogation of Mafa phosphorylation leading to an excess of Mafa protein in Mafa^4A/4A mutants has a dramatic consequence on survival at birth.

**Mafa-4A mutation in the central nervous system causes neonatal apneic breathing**. When the Mafa^flox4A/flox4A line was crossed with a nestin Cre line (Fig. 1a), no more than 10% (3/34 expected) conditional nestin^Cre/+; Mafa^flox4A/flox4A mutants (hereafter designated Mafa^n4A/n4A), survived up to weaning stage (Supplementary Table 3). Most Mafa^n4A/n4A newborns, like Mafa^4A/4A ones, were found hypoactive, developed cyanosis and died between P0 + 12 h and P2 (Supplementary Fig. 2a) indicating that the Mafa apneic phenotype is CNS specific.

We then investigated the ventilation of Mafa^n4A/n4A mice at birth. Non-invasive unrestrained whole-body plethysmography was performed on control (n = 29), and Mafa^n4A/n4A (n = 23) mutant pups over the P0-P2 period. At birth, all littermates established within 1 h a regular breathing pattern, which then rapidly deteriorated in mutants via an abnormal increase in the incidence of respiratory pauses (Fig. 1a). Twelve hours after birth (P0 + 12 h), Mafa^n4A/n4A mutant pups presented with an approximately halved minute volume ventilation ($V_M$) compared to control littermates, which was accounted for by an abnormally high incidence of apneic events that lowered the breathing frequency ($F_R$) while tidal volumes ($V_T$) were maintained normal (Supplementary Fig. 2b). To quantify the incidence of apneic events (i.e., respiratory cycles longer than normal), we statistically

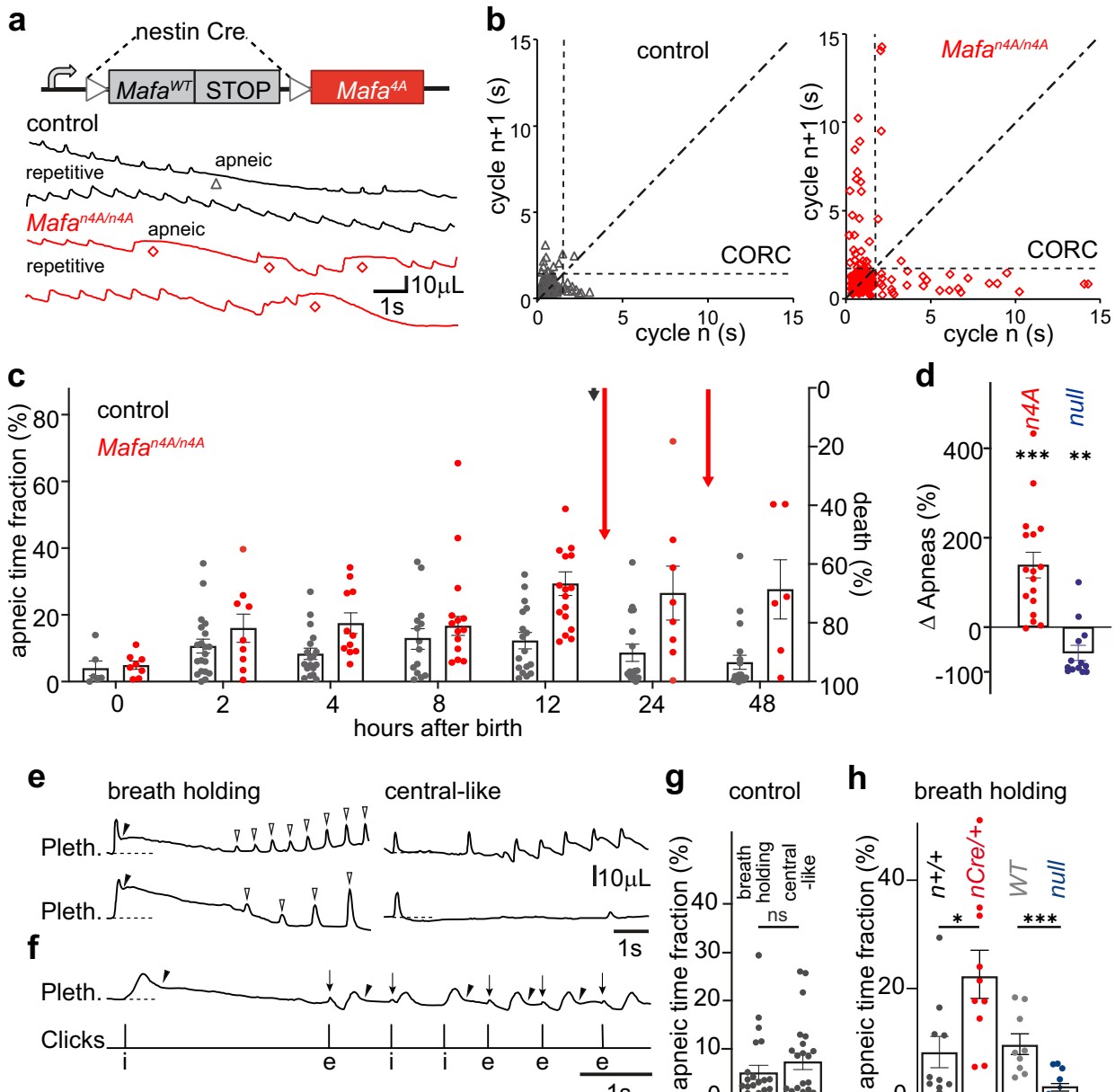

**Fig. 1 Impaired Mafa phosphorylation in the nervous system results in lethal breath holding apneas. a** Top, genetic scheme to introduce the Mafa4A mutation in neurons. Bottom, typical plethysmographic recordings (inspiration: upward) of control (black traces) and $Mafa^{n4A/n4A}$ (red traces) littermates at P0 + 12 h showing repetitive and apneic breathing cycles (the superposed traces are adjacent segments of a continuous record). Apneas (gray triangle in controls) are more frequent in the mutant (red diamonds). **b** Poincaré plots of the duration of consecutive respiratory cycles $n+1$ (ordinates) as a function of $n$ (abscissae) in a control (left) and a $Mafa^{4A/4A}$ (right, 5 min recording samples). Repetitive short cycles (filled symbols) are clustered on the diagonal below the CORC (vertical and horizontal dotted lines), apneic cycles (empty symbols) give points away from the diagonal beyond the CORC. Note that the $Mafa^{n4A/n4A}$ mutant shows an increased incidence of apneas. **c** Evolution of the apneic time fraction (left ordinates axis, bar plot) and mortality (right inverted ordinates axis, vertical arrows) in control (gray, $n = 29$) and $Mafa^{n4A/n4A}$ mutants (red, $n = 21$) pups. Each symbol represents measurement of a single pup (One-way ANOVA between the two genotypes $P < 0.0001$, degree of freedom (df) = 1, $F = 34.38$.). **d** Percent relative change in the number of apneas (ordinates) compared to their controls for $Mafa^{n4A/n4A}$ (Statistical test between mutant and control littermates: n4A vs $Cre^{+/+}$ littermates $n = 17$ for both genotypes, two-sided unpaired $t$-test $P = 0.0004$, df = 32, $t = 3.965$) and $Mafa^{LacZ/LacZ}$ (null $n = 13$ vs wild type $n = 10$ littermates, two-sided Mann–Whitney test $P = 0.0041$, $U = 20$). **e** Example traces of the two types of apneas in $Mafa^{n4A/n4A}$ mutant pups. At left, breath holding apneas characterized by post-inspiratory onset (black arrowheads) of limited expiratory flow prolonging lung inflation (plateau above dotted baseline) followed by the resuming of inspiratory efforts of progressively increasing amplitudes (empty arrowheads). At right, central-like apneas characterized by normal inspiratory and expiratory phases separated by an abnormally long pause during which the lung is deflated. **f** Joint plethysmographic (top trace) and click audio recordings (bottom trace) during an epoch of breath holding apneas. Clicks that follow a post-inspiratory (arrowheads) breath holding apnea are associated to small upward pressure shifts (downward arrows) that immediately precede onset of an expiration (e below the click trace, 4 of 5 clicks) deflating the lung or occasionally an inspiration (i below the click trace, 1 of 5 clicks,) further inflating the lung. **g** Quantification of the apneic time fraction for apnea types in control pups ($Mafa^{flox4A/4A}$, $n = 23$) at P0 + 12 h (two-sided Mann–Whitney test $P = 0.4100$, $U = 226.5$). **h** Quantification of the breath holding apneic time fraction in $Mafa^{n4A/n4A}$ (nCre/+, $n = 10$ pups, red), $Mafa$ null ($n = 13$ pups, blue) and that of respective control (n+/+, $n = 10$ pups, gray, two-sided unpaired $t$-test $P = 0.0145$, df = 18, $t = 2.704$) and wild type (light gray, $n = 9$ pups, two-sided Mann–Whitney test $P = 0.0052$, $U = 14$) littermates. Plots represent mean ± sem.

analyzed respiratory cycles in each breathing record, and defined a "Cut-Off Respiratory Cycle" duration (CORC) beyond which respiratory cycles were considered apneic (see Methods). Respiratory cycles were subdivided into two groups, (i) repetitive short cycles corresponding to regular breathing (i.e., followed and preceded by a similar short cycle; "repetitive" in Fig. 1a), yielding a low value cluster on the diagonal of Poincaré plots (filled symbols, Fig. 1b) and (ii) isolated cycles longer than the CORC (i.e., preceded and followed by a short cycle; "apneic" in Fig. 1a) yielding the values distributed away from the diagonal of Poincaré plots (empty symbols, Fig. 1b). With this tool in hand, we analyzed the temporal emergence of apneas in mutant and control neonates by comparing the percentage of time spent in apnea (apneic time fraction) during periods of quiet breathing (Fig. 1c). In control mice, we observed a progressive increase in the apneic time fraction over the P0 + 2–12 h period after which the incidence of apneic events gradually decreased. By contrast, $Mafa^{n4A/n4A}$ mutant neonates displayed exaggerated apneic breathing compared to controls that was first evident at P0 + 4 h and which worsened in frequency (Supplementary Fig. 2c) and duration (Supplementary Fig. 2d) to culminate in an apneic time fraction twice that of control littermates at P0 + 12 h ($Mafa^{n4A/n4A}$: 29.3 ± 3.5%, $n = 17$ vs control: 12.3 ± 2.5%, $n = 17$, Fig. 1c). Whereas all $Mafa^{n4A/n4A}$ mutant pups survived until P0 + 12 h, they began to die thereafter (Fig. 1c, red vertical arrows). Therefore, an apneic time fraction of less than 15% as observed in control pups, appears to be physiologically acceptable (at least for some hours), while its increase over 25%, as in the mutants, compromises survival. Intriguingly, in contrast to $Mafa^{n4A/n4A}$ mutants, $Mafa$ null mutants showed a reduced incidence of apneas when compared to controls, further confirming the pro-apneic action of Mafa and suggesting a gain-of-function mutation (Fig. 1d).

Plethysmographic traces of the breathing patterns of control and $Mafa^{n4A/n4A}$ mutant pups displayed two types of apneic events. The first type was characterized by an abrupt post-inspiratory reduction of expiratory flow that prolonged lung inflation, suggesting a closure of the airways, and that we termed "breath holding apneas", the second type consisted of a respiratory pause at the end of expiration (lung deflated) and was reminiscent of central apneas (Fig. 1e). In breath holding apneas, the slow deflation of the lung could be accompanied by a resuming of rhythmic inflation efforts that progressively, on a cycle-to-cycle basis, showed increased tidal volumes and likely reflected the progressive re-opening of the airways (open arrowheads in Fig. 1e) or could be terminated by abrupt expiratory-like lung deflations (Fig. 1f). To gain indirect insights on the status of airways during breath holds, we monitored audio correlates of upper airway function (see Methods), more precisely, we focused on brief (ms order) broad-band audio-mechanical events named "clicks" or cracking sounds (see refs. [9,30,31]) that critically rely on openings of airway cavities from a closed state[32]. Indeed, clicks have been reported to often associate with ultrasonic vocalizations (USVs)[30,33] and, more precisely, were found to be temporally locked to lung post-compressions' re-openings of closed airways powering calls during vocal breathing[9] (Supplementary Fig. 3a). We found that like vocalizations, breath holding apneas were accompanied by clicks (69.5% fraction of breath holds with click, $n = 46$ breath holds, from 9 wildtype pups, Fig. 1f) which were systematically time locked to small amplitude upward pressure shifts of the plethysmographic trace (downward arrows in Fig. 1f and Supplementary Fig. 3b) that immediately precede expiratory-like lung deflations terminating breath holds. Interestingly, clicks were virtually absent during both (non-vocal) eupneic breathing (Supplementary Fig. 3c) and central-like apneas (Supplementary

Fig. 3d, e). Although a direct measurement of airway patency was not feasible in our experimental conditions, these data strongly support the obstructive nature of breath holding apneas in mutant pups.

Breath holding and central-like types of apnea contributed similarly (about 5%) to the apneic time fraction of control ($Mafa^{flox4A/flox4A}$) pups at P0 + 12 h (Fig. 1g). In $Mafa^{n4A/n4A}$ mutants at the same stage, the incidence of "breath holding apneas" was selectively increased about three-fold ($n+/+$, 8.3 ± 2.8%, $n = 10$ vs. $nCre/+$, 22.6 ± 4.4%, $n = 10$). Conversely, in $Mafa^{LacZ/LacZ}$ null mutants, breath holding apneas were more than five-fold reduced ($Mafa^{+/+}$, 9.7 ± 1.9%, $n = 9$ vs $Mafa^{LacZ/LacZ}$, 1.7 ± 0.7 %, $n = 13$, Fig. 1h). Altogether these data indicate that the incidence, specifically, of breath holding apneas is commensurate with the rates of Mafa protein in central neurons, pointing to an unexpected link between Mafa post-translational regulation and the control of airway patency at birth.

### Mafa expression in the central nervous system includes inhibitory neurons in the caudal medulla. We looked for candidate neurons causal to the respiratory deficit by examining Mafa-expressing (Mafa+) neuronal populations using the $Mafa^{LacZ/+}$ line. Mafa+ neurons were mostly found in sensory regions of the central and peripheral nervous system: the olfactory bulb, auditory networks of the pons, the spinal trigeminal nucleus, and, as previously shown, in dorsal root ganglia and in ventral and dorsal neurons of the spinal cord[25–27,34,35] (Fig. 2a). None of these Mafa+ neuronal populations are known to participate in the control of breathing. Conversely, two regions with a prominent respiratory role, the preBötzinger complex (Fig. 2b) that generates the respiratory rhythm (see Ref. [36] for a review) and the retro-trapezoid nucleus (Fig. 2c) that modulates it as a function of $CO_2$ levels (see Ref. [37] for a review) were devoid of Mafa+ neurons. Accordingly, comparable rhythmic inspiratory-like activity was recorded from the fourth cervical spinal root in isolated brainstem preparations from $Mafa^{n4A/n4A}$ and control P0 littermates (Supplementary Fig. 4a, b), and $Mafa^{n4A/n4A}$ mutant pups had a preserved $CO_2$ chemoreflex at birth (Supplementary Fig. 4c). Taken together, these results show that the respiratory distress of $Mafa^{n4A/n4A}$ mutants is caused by alterations outside of the preBötC rhythm generator and of its attendant RTN modulator, in keeping with the notion that $Mafa^{n4A/n4A}$ mutants present with breath holding, rather than central-like, type apneas.

Interestingly, Mafa was interestingly expressed in two discrete populations of the medulla reticular formation, one flanking the lateral and ventral margins of the hypoglossal motor nucleus (Mo12) here termed the peri-hypoglossal area (peri12) including its accessory nucleus (accMo12, Fig. 2d, e, h), and another, more caudal population located close to the nucleus ambiguus (peri-nAmb, Fig. 2f, g, i). In situ hybridization indicated that Mafa+ neurons in both of these regions were predominantly inhibitory (83% Slc6a5+ (Glyt2+), 76% Gad1+, 72% Gad2+ compared to 7% Slc17a6+ (VGlut2+), Fig. 2j).

Taken together, these data suggest that the 4A mutation might enhance the incidence of breath holding apneas through an action on medullary inhibitory neurons.

### The Mafa 4A mutation acts on GABAergic transmission. To examine the possibility that the respiratory deficit of $Mafa^{n4A/n4A}$ mutants may be related to inhibitory synaptic transmission, we injected newborn pups subcutaneously with pentylenetetrazol (PTZ), a GABA$_A$-receptor antagonist that crosses the blood brain barrier. Using a PTZ dose (40 μg/g) that does not induce seizures in neonates[38], we compared the ventilation of PTZ treated $Mafa^{n4A/n4A}$ and control pups at the P0 + 12 h time, when the

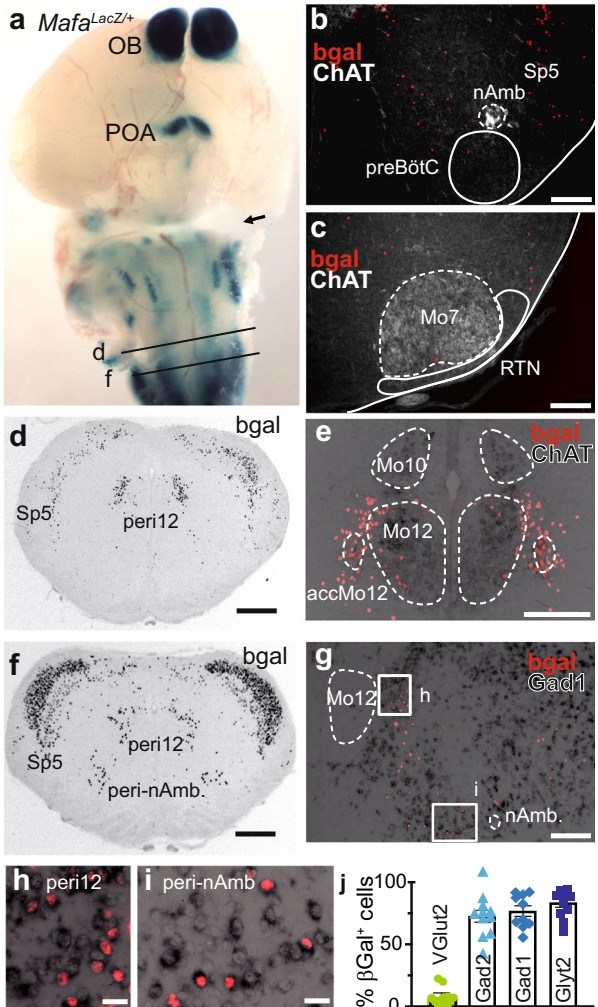

**Fig. 2 Mafa+ neurons of the reticular formation in the caudal medulla are inhibitory. a** Whole mount (ventral view) of a *Mafa* $^{LacZ/+}$ brain at E15 showing *Mafa* expression (Xgal staining, dark blue) restricted to the olfactory bulb (OB), to the pre-optic area of the hypothalamus (POA) to the brainstem (pons and medulla) and to the spinal cord. **b** Hemi-transverse section (medial left, dorsal top) of a *Mafa* $^{LacZ/+}$ pup at P0 showing that the preBötC area, below ChAT+ neurons (white) of the nucleus ambiguous (nAmb), is devoid of gal+ cells (red). **c** Same as b showing the absence of bgal+ cells (red) in the RTN, below ChAT+ neurons (white) of the facial motor nucleus (Mo7). **d** Transverse section of a *Mafa* $^{LacZ/+}$ pup at P0 (at the d axial position indicated in a) showing Mafa+ cells immunostained for bgal (black) in the caudal medulla in the spinal part of the trigeminal nucleus (sp5), in the peri12 reticular formation laterally to the hypoglossal motor nucleus (Mo12), in motoneurons of the accessory hypoglossal motor nucleus (acc-Mo12). **e** Close-up view of Mafa+ peri12 neurons (red) counterstained for ChAT+ (black) motoneurons at the same axial level as **d**. Transverse section (at the **f** axial position indicated in **a**) showing Mafa+ cells in the Sp5 and medial to the nucleus ambiguus in the peri-nAmb area. **g** Close up view of a hemi-transverse section (medial left, dorsal top) showing Mafa+ cells (red labeling) combined to Gad1 in situ hybridization (black) at the same axial level as (**f**) showing Gad1+/Mafa+ cells in the peri12 and peri-nAmb. Close-up view from the (**h**) inset in (**g**). Close-up view from the **i** inset in (**g**). **j** Quantification of neurochemical types of Mafa+ neurons of the caudal medulla reticular formation by in situ hybridization for *VGlut2* (13 section counts), *Glyt2* (11 section counts), *Gad1* (10 section counts) and *Gad2* (12 section counts). Note that Mafa+ neurons of the caudal reticular formation (peri12 and peri-nAmb regions) are predominantly inhibitory. Plots represent mean ± sem. Scale bars (μm): 100 (**b–g**), 20 (**h**, **i**).

incidence of apnea is maximal. After administration of PTZ, the regular breathing frequency of both control and mutant pups was found increased (Fig. 3a). In addition, PTZ injection, but not saline, reduced by more than 40% the apneic time fraction of *Mafa* $^{n4A/n4A}$ mutant pups but had no effect on that of controls (Fig. 3b). The PTZ-induced rescue of *Mafa* $^{4A/4A}$ mutants apneic breathing was transient (0.5–1 h) in line with the short half-life of PTZ[39]. None of the mutants (0/10) died within the one-hour period following PTZ injection, with 5/10 then dying in the next following two-hour period, when PTZ was no longer effective. Notably, PTZ reduced the apneic time fraction through a selective effect on apneas of the "breath holding" type (Fig. 3c) suggesting that these are caused by endogenously over-active inhibitory GABAergic circuits.

To further examine the role of inhibitory neurons in the 4A mutation-induced respiratory deficit and specifically in promoting breath holding apneas, we conditionally knocked in the 4A mutation in inhibitory neurons using a *Slc32A1-ires-Cre* line[40] (*VGAT* $^{Cre}$, Fig. 3d). VGAT is a vesicular inhibitory aminoacid transporter expressed in both GABAergic and glycinergic neurons. We crossed *VGAT* $^{Cre/+}$; *Mafa* $^{flox4A/+}$ with *Mafa* $^{flox4A/flox4A}$ mice and recorded the ventilation of *VGAT* $^{Cre}$; *Mafa* $^{flox4A/flox4A}$ (*Mafa* $^{VGAT4A}$) mutant neonates. Fifty five percent (6/11) of *Mafa* $^{VGAT4A}$ mutant pups died before P2 compared to only 15% (2/13) death observed among control pups. Moreover, plethysmographic recordings indicated that *Mafa* $^{VGAT4A}$ mutants showed an apneic time fraction double that of control littermates (Fig. 3e–g) accounted by a selective increase in the incidence of breath holding apneas (Fig. 3h). The reasons why *Mafa* $^{VGAT4A}$ mutants show reduced morbidity compared to *Mafa* $^{n4A/n4A}$ mutant is presently unknown. These data show that restricting the 4A mutation to inhibitory neurons was sufficient to recapitulate the severe increase in the incidence of breath holding apneic events.

**Mafa is expressed by premotor neurons controlling upper airway patency.** We then refined the search for candidate Mafa+ inhibitory hindbrain neurons to examine whether breath holding apneas may result from (i) altered sensory control of breathing originating in trigeminal and vagally-derived sensory neurons that innervate the larynx and the lungs and mediate powerful protective reflexes to slow down or pause breathing[41,42] and/or (ii) altered GABAergic transmission in dorsolateral pontine respiratory areas as the case in a mouse model of the Rett syndrome[43]. Using RNAscope, we report that the jugular and nodose vagal ganglia are largely devoid of Mafa+ neurons (Fig. 4a) and that the few present, as the case in the trigeminal sensory ganglia (Fig. 4b), co-expressed VGlut2 and are thus excitatory (Fig. 4c). Next, using the intersectional line *VGAT* $^{Cre/+}$; *Mafa* $^{floxLacZ/+}$ (*Mafa* $^{VGATLacZ}$) to selectively label Mafa+ inhibitory neurons we found that they were also absent from the secondary sensory structures targeted by the above sensory afferents, namely the paratrigeminal nucleus (Pa5), the principal trigeminal nucleus (Pr5) and in the nucleus of the solitary tract (nTS). Furthermore, pontine dorsal respiratory areas comprising the Kölliker-Fuse and the parabrachial nucleus also lacked Mafa+ inhibitory neurons (Fig. 4d). These findings argue strongly that Mafa mutant apneic breathing is unlikely to be caused by alterations of respiratory sensory feedbacks or sensory integration at medullary of pontine levels.

By contrast, in keeping with the probable effect of the 4A mutation on: (i) upper airway motor control and (ii) the presence of inhibitory candidate neurons in the medulla reticular formation we then looked for evidence that the *Mafa* $^{4A}$ mutation targets neurons in the motor arm of respiratory control. We

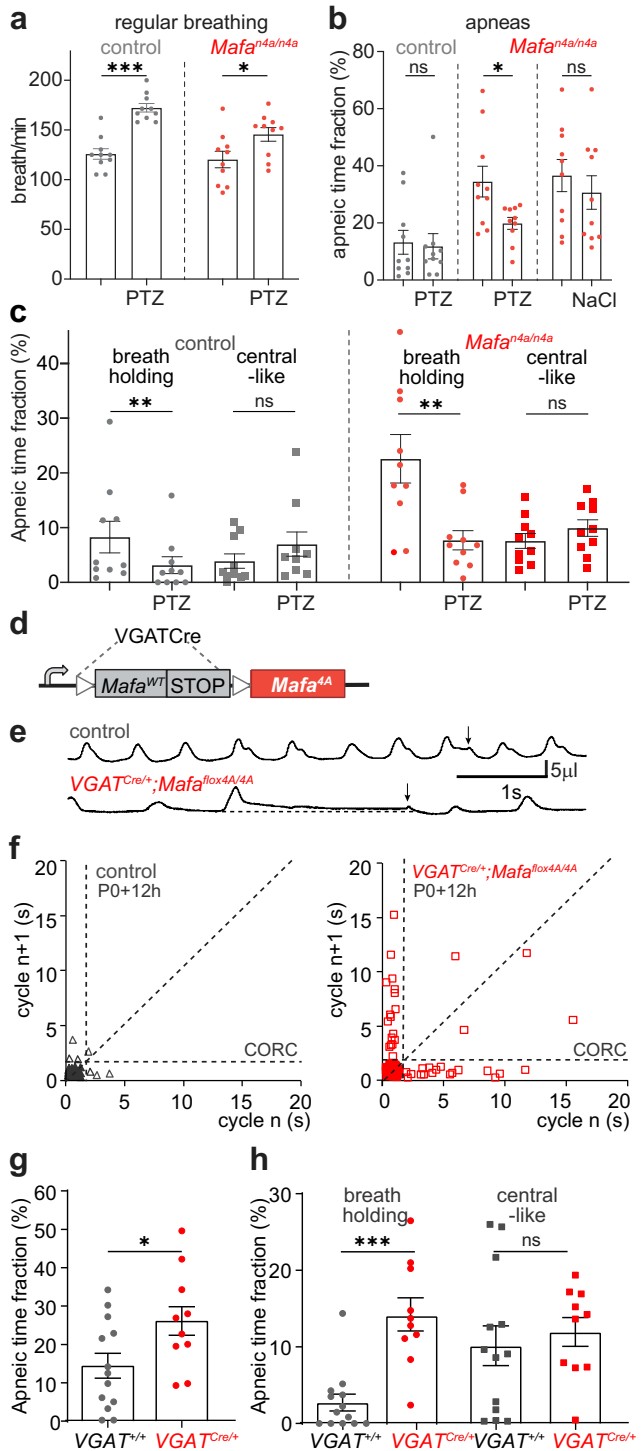

**Fig. 3 Mafa-4A-induced breath holding apneas rely on GABAergic synaptic transmission. a** The GABA$_A$-receptor antagonist PTZ increases regular breathing rates in control (gray symbols, $n = 10$, two-sided paired Student's $t$ test, $P < 0.0001$, df = 9, $t = 6.851$) and $Mafa^{n4A/n4A}$ (red symbols, $n = 10$, two-sided paired Student's $t$ test $P = 0.040$, df = 9, $t = 2.393$) at P0 + 12 h (Student's $t$ test). **b** PTZ has no effect on the apneic time fraction of control pups ($n = 10$, gray symbols, two-sided Wilcoxon paired test, $P = 0.921$) but PTZ and not NaCl reduces the apneic time fraction of $Mafa^{n4A/n4A}$ mutant pups ($n = 10$, red symbols, two-way ANOVA $F(1,18) = 8.176$, $P = 0.0104$ followed by Bonferroni's multiple comparison test, $P = 0.0202$ for PTZ; $P = 0.5147$ for NaCl). **c** PTZ reduces breath holding apneas in control ($n = 10$ pups, two-sided Wilcoxon test $P = 0.0039$ and $Mafa^{n4A/n4A}$ mutant pups ($n = 10$, two-sided paired Student's $t$ test $P = 0,0098$, df = 9, $t = 3.264$) but not central-like apneas of control (two-sided Wilcoxon test $P = 0.0645$) nor of $Mafa^{n4A/n4A}$ mutant pups (paired Student's $t$-test $P = 0.2791$, df = 9, $t = 1.152$). **d** Genetic recombination to target the 4A mutation in inhibitory neurons. **e** Plethysmographic recordings of control (upper trace) and of $VGAT^{Mafa4A}$ mutant pups (lower trace) at P0 + 12 h showing apneic breathing and upwards pressure shifts (arrows) terminating breath holding apneas. **f** Poincaré plots of a control (left, black symbols) and of a $Mafa^{VGAT4A}$ mutant presenting with increased number of apneas (red symbols beyond the CORC). **g** The apneic time fraction of $Mafa^{VGAT4A}$ pups ($n = 10$) is about double that of control littermates ($n = 13$, two-sided unpaired $t$-test $P = 0.0263$, df = 22, $t = 2.382$). **h** The apneic time fraction corresponding to breath holding apnea is three-fold higher in $Mafa^{VGAT4A}$ ($VGAT^{Cre/+}$, $n = 10$, red symbols) than controls ($VGAT^{+/+}$, $n = 13$, gray symbols, two-sided Mann–Whitney test $P = 0.002$, $U = 10$) while that corresponding to central-like apneas is not changed by the mutation (unpaired $t$-test $P = 0.5774$, df = 20, $t = 0.5663$). Plots represent mean ± sem.

in the dorsal part of the intermediate reticular formation in overlap with the peri12 group of Mafa$^+$ inhibitory neurons (Fig. 4f). There, we found that Mafa$^+$/mCherry$^+$ virally labeled geniohyoid premotor neurons (Fig. 4g, h) accounted for $10.0 \pm 0.4\%$ of all geniohyoid premotor neurons of the peri12 area (counted from 1123 mCherry$^+$/GFP$^-$ labeled neurons in 4 pups). These data reveal the existence of a Mafa$^+$ inhibitory premotor neuronal subset in a position to depress the motor drive of the geniohyoid muscles, thus facilitate the collapse of upper airways. We next sought the mechanism whereby Mafa expression could increase inhibitory neuronal activity.

**Mafa-4A mutation does not alter neuronal development**. As *Mafa* is expressed early in neural progenitors[28], we investigated the possibility that *Mafa* mutations may interfere with the development of inhibitory neurons. Focusing on neurons of the medulla which form an exclusive Mafa$^+$ inhibitory population we have compared their number across the different *Mafa* mutant genotypes at P0 (Supplementary Fig. 5). Since the Mafa protein is not detectable by immunostainings in wild type animals, but only in 4A homozygous mutants where the protein is stabilized, we compared ($n = 10$ sections, from 2 animals/genotype) the number of Mafa$^+$ neurons (immunostained for Mafa) in $Mafa^{4A/4A}$ to that of $Mafa^{LacZ/+}$, $Mafa^{LacZ/4A}$ and $Mafa^{LacZ/LacZ}$ (immunostained for βgal). No significant differences in the number of Mafa$^+$ neurons were detected between the different genotypes. These data indicate that Mafa affects neither the proliferation, nor the survival or the migration of inhibitory neurons in which it is expressed but, leave open the possibility that it may potentiate their synaptic efficacy.

***Gad2* is a direct Mafa target gene**. One way in which Mafa could regulate GABAergic transmission is by controlling GABA

therefore searched for Mafa$^+$ inhibitory neurons in the circuit motorizing an upper airway opener muscle by trans-synaptic viral tracing[44]. To do this, we injected unilaterally, in $Mafa^{LacZ/+}$ pups at P3, a G-defective rabies virus variant encoding the fluorophore mCherry together with a helper virus encoding G and the fluorophore YFP in the geniohyoid muscle (i.e., that pulls the hyoid bone forward to increase airway patency) (Fig. 4e) and examined their brains on P9. As expected, the only seed neurons (i.e., that co-express both fluorophores) were found in the accessory hypoglossal motor nucleus[45] while the bulk of premotor neurons, presynaptic to the seed motoneurons (i.e., that express only the rabies virus encoded mCherry) was located

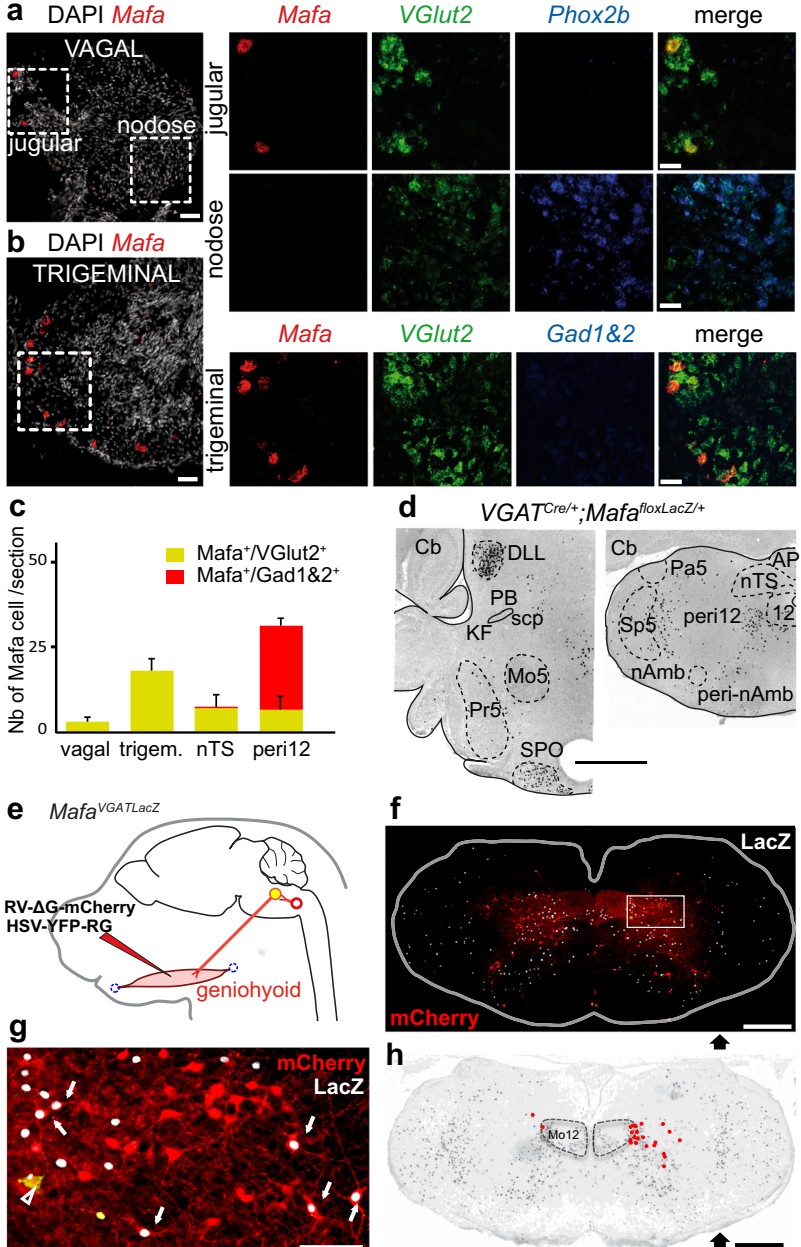

**Fig. 4 Absence of Mafa inhibitory neurons in sensory and pontine respiratory structures and presence of Mafa inhibitory premotor neurons controlling the geniohyoid muscle. a** Left, transverse sections of P0 wild type pups showing the vagal jugular and nodose ganglia stained for DAPI (white) and by RNAscope using a probe against *Mafa* (red). Right, top, close-up views from the jugular ganglion probing *Mafa* (red), *VGlut2* (green), *Phox2b* (blue) and the merge. Right, bottom, close-up views from the nodose ganglion. **b** Transverse sections of the trigeminal ganglion stained for DAPI (white) and *Mafa* (red). Right, close up views of the inset in (**b**) probing *Mafa (red)*, *VGlut2 (green)* and *Gad1&Gad2* (blue). Note the absence of *Mafa* expression in the nodose ganglion, expression of *Mafa* in the jugular (sparse) and trigeminal ganglia restricted to *VGlut2*-expressing excitatory neurons. **c** Summary of RNAscope probing for co-expression of *Mafa/VGlut2*- and *Mafa/Gad1&2*-expression in vagal (5 sections, $n = 3$ ganglia) and trigeminal (7 sections, $n = 3$ ganglia) ganglia, nucleus tractus solitarius (nTS, 8 sections, $n = 2$ pups) and peri12 (8 sections, $n = 2$ pups). Plots represent mean ± sem. **d** Left, transverse hemi-section of the brainstem at pontine level of a *VGAT*$^{Cre/+}$; *Mafa*$^{FloxLacZ/+}$ P8 pup, showing absent Mafa$^+$ inhibitory neurons (black) in the PB/KF, Pr5 areas but their presence in the SPO and DLL. Right, representative transverse hemi-section of the brainstem at medullary level showing the presence of Mafa$^+$ inhibitory neurons in the peri12, peri-nAmb but their virtual absence in the Sp5, Pa5 and nTS ($n = 2$ pups). **e** Monosynaptic tracing scheme showing unilateral injection of G-deleted Rabies viruses encoding mCherry and of a helper G- and YFP-encoding HSV virus into the geniohyoid muscle to transynaptically trace the position of premotor neurons. **f** Representative transverse section (from $n = 4$ *VGAT*$^{cre/+}$; *Mafa*$^{floxLacZ/+}$pups) immunostained for βgal (white) in the caudal medulla showing mCherry$^+$ virally labeled premotor neurons (red) and rare (at this axial level) seeding mCherry$^+$/YFP$^+$ geniohyoid motoneurons (yellow) in the accessory hypoglossal motor nucleus (arrowhead). **g** Close-up view of the inset in **f** showing that a fraction of mCherry$^+$ premotor neurons indicated by arrows are Mafa$^+$ (mCherry$^+$/βgal$^+$, white nuclear labeling). **h** 2D-reconstruction of the position of Mafa$^+$ geniohyoid premotor neurons (red dots) and Mafa$^+$ cells (gray dots) showing their location in the peri12 area. DLL dorsal nucleus of the lateral lemniscus, KF Kölliker-Fuse nucleus, PB parabrachial nuclei, Pa5 paratrigeminal nucleus, Pr5 principal sensory trigeminal nucleus, scp superior cerebellar peduncle, SPO superior paraolivary nucleus. Scale bars (μm): 100 (**a**, **b** left, **c**), 50 (**a**, **b** right), 1000 (**d**), 500 (**f**, **h**), 100 (inset).

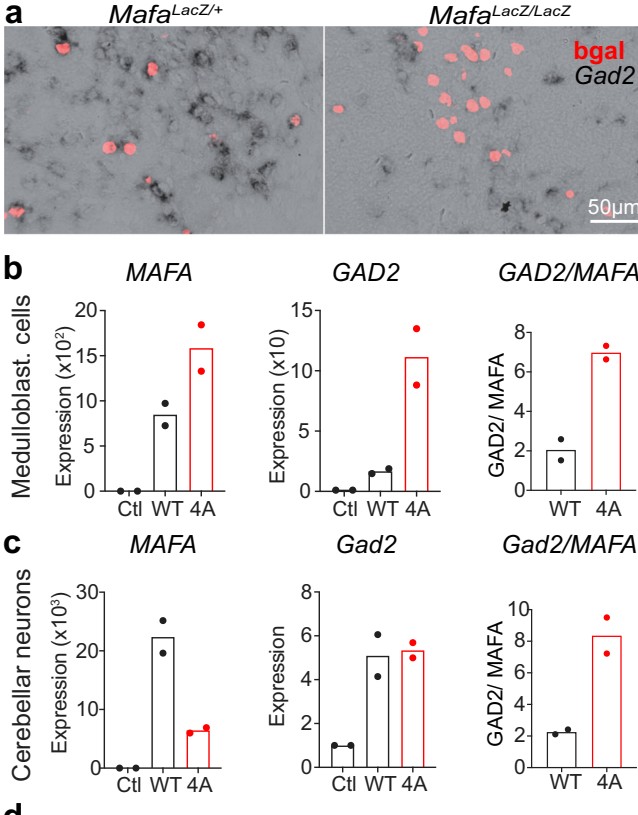

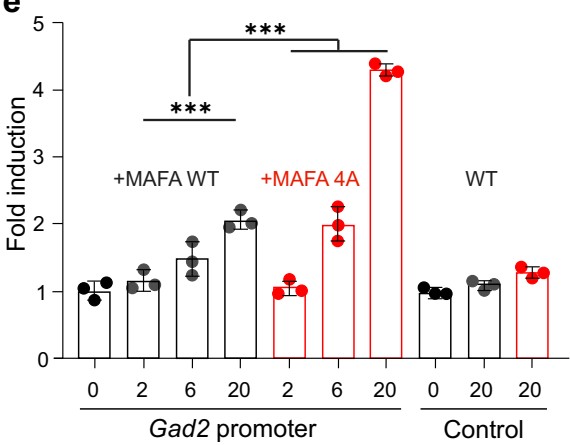

**Fig. 5 *Gad2* is a direct target of the Mafa transcription factor. a** *Gad2* in situ hybridization (black) combined with anti-bgal staining (red) on P0 transverse sections of the caudal medulla of section of *Mafa*$^{LacZ/+}$ (left) and *Mafa*$^{LacZ/LacZ}$ (right) pups. Note the reduced *Gad2* signal surrounding βgal$^+$ nuclei in the absence of Mafa. Data from three mice for each genotype. In DAOY medulloblastoma cell line (**b**) or primary culture of granule cell progenitors (**c**) that do not express *MAFA* (Ctl, left plots in **b** and **c**), transfection with either MAFA WT or MAFA 4A yields *MAFA* expression (left plots) that induces *GAD2* transcription (middle plots). The right panel shows the fold induction of *GAD2*, compared to *MAFA* expression ($n = 2$ biological replicates each with $n = 3$ technical replicates). **d** The *Gad2* promoter (878nt upstream of *Gad2* transcription start, in between oligos used to clone *Gad2* promoter in pGL3) contains MAF Responsive Elements (MARE, red boxes). **e** Luciferase assays in DAOY cells reporting *Gad2* promoter (left) and control luciferase activity (right) when co-transfected with increasing doses of plasmids encoding MAFA WT or MAFA 4A. All assays were performed in triplicates. Two-way ANOVA was used to test that MAFA enhances *Gad2* promoter activity in a dose-dependent manner ($P < 0.0001$, df $= 3$; $F = 218.4$) and that MAFA 4A induces endogenous *GAD2* transcription more efficiently than MAFA WT ($P < 0.0001$, df $= 1$; $F = 93.8$). Plots represent mean ± sem. Scale bar (μm): 50 (**a**).

(medulloblastoma, DAOY) that do not express *MAFA*. As measured by RT-qPCR, Mafa significantly increased *GAD2* expression in both cell lines (Fig. 5b) with *GAD2* expression being four times higher in Mafa-4A expressing cells. These results were further confirmed on primary neuronal cells using mouse cerebellum primordia cultures (Fig. 5c). Inspection of the *Gad2* proximal promoter region revealed the presence of at least 3 potential Maf Responsive Element sequences (MARE[46], Fig. 5d). A *Gad2* promoter region containing these 3 potential MAREs was then cloned into the pGL3 Luciferase reporter vector (Gad2-Luc) and a medulloblastoma cell line was transfected with either control pGL3 (containing a minimal promoter without MAREs) or the Gad2-Luc vectors in the presence of increasing concentrations of plasmids encoding WT or Mafa-4A protein. WT Mafa was found to activate the *Gad2* promoter in a dose-dependent manner, with the Mafa-4A mutant protein being two times more potent (Fig. 5e). These data demonstrate that *Gad2* is a direct Mafa target gene and that GSK3-mediated phosphorylation of Mafa likely attenuates GABAergic synaptic transmission via a mechanism that lead to down control of *Gad2* transcription.

**Chemogenic activation of Mafa$^+$ inhibitory neurons triggers apneas.** To confirm that the *Mafa*$^{n4A/n4A}$ apneic phenotype arises from increased activity of Mafa$^+$ inhibitory neurons, we used an intersectional chemogenetic approach. We first verified that Mafa$^+$ inhibitory neurons could be targeted by dual recombination with a *Mafa*$^{Flpo}$ and a *VGAT*$^{Cre}$ acting on the *Rosa26*$^{FeLa}$ reporter line[47] (*VGAT*$^{Cre/+}$; *Mafa*$^{Flpo/+}$; *RC::Fela/+*), which specifically labels Mafa$^+$/VGAT$^-$ (excitatory) neurons with GFP and Mafa$^+$/VGAT$^+$ (inhibitory) neurons with βgal (Supplementary Fig. 6a–d). Furthermore, we confirmed that Mafa$^+$ neurons targeted in the caudal medulla were almost exclusively inhibitory. In addition to the cell groups detected by ISH for *Mafa* in wildtype animals or by βgal immunostain in *Mafa*$^{LacZ/+}$ mutants, this background surprisingly revealed doubly recombined neurons in the area of the preBötC respiratory rhythm generator, that at birth is devoid of Mafa expression (Fig. 2; Supplementary Fig. 6e) and functional in *Mafa*$^{n4A/n4A}$ mutants (Supplementary Fig. 4). This is a likely consequence of an early and transient phase of Mafa expression in the rhombencephalon

synthesis. We first confirmed qualitatively by ISH that inhibitory Mafa$^+$ neurons in the medulla in *Mafa null* mutants, expressed lower amounts of *Gad2* transcripts compared to heterozygous *Mafa*$^{LacZ/+}$ littermates (Fig. 5a). We next performed RT-qPCR to assess quantitatively the hypothesis that in *Mafa*$^{4A/4A}$ mutants, the higher rates of Mafa would result in augmented Gad2 synthesis. To do this, plasmids encoding wildtype (WT) or Mafa-4A were transfected into a human neuronal cell lines

(Supplementary Fig. 6f, g). With this in mind, we used the dual Mafa/VGAT recombinogenic background to trigger expression of a conditional G-protein coupled receptor hM3Dq-mCherry fusion DREADD (Designer Receptor Exclusively Activated by a Designer Drug) in all these neurons (Fig. 6a, Supplementary Fig. 6h). The breathing behavior of 12 intersectional DREADD mutant neonates (P1-P2) was monitored before and after a single subcutaneous injection of the DREADD agonist clozapine N-oxide (CNO, 5 mg/kg). Ten of twelve experimental pups displayed significantly increased apneas after CNO treatment (Fig. 6b, c). The incidence of apneas started to increase 10–20 min after CNO injection, reaching a plateau during which the pups presented on average with a 25% apneic time fraction. These experimental mice recovered normal breathing two hours after the CNO injection except for 2 that could not reverse the induced apneic breathing and died shortly after the end of recordings. No increase in apnea or lethality was observed following CNO injection in control littermates ($VGAT^{+/+}$; $Mafa^{Flpo/+}$;$RC$::$FL$-$hM3Dq/+$, Fig. 6d) and no effect of saline vehicle was observed on mutants (Fig. 6e). CNO treatment increased the incidence of breath holding apneas in keeping with preceding results following knock-in of the 4A mutation and PTZ pharmacological treatment. However, CNO treatment also augmented the incidence of central-like apneas (Fig. 6f) possibly through the spurious access to preBötC neurons revealed above in the lineage tracing analysis. We conclude that acute activation of Mafa+ inhibitory neurons in vivo is able to trigger breath holding apneas thus mimicking the breath holding phenotype of $Mafa^{n4A/n4A}$ mice at birth.

## Discussion

Apnea remains one of the major respiratory disorders spanning all ages from infants to adults. We found in mouse neonates a relationship between the stabilization of the transcription factor Mafa (via its 4A mutation) and an increased incidence of breath holding apneas incompatible with survival. We show that $Gad2$ is a target gene of Mafa; that the apneic phenotype can be rescued by blocking $GABA_A$ receptor-mediated synaptic transmission and that apneas are promoted by chemogenetic activation of Mafa-expressing inhibitory neurons or by their selective expression of the 4A mutation. Our analysis of the ventilation of $Mafa$ mutant neonates has revealed a spectacular link between $Mafa$ and the incidence of apneas, particularly breath holding apneas. When compared to wild types, $Mafa$ null mutants devoid of Mafa, and 4A mutants with higher than normal rates of Mafa, showed reduced and increased incidence of apneas, respectively, suggesting that Mafa transcription factor rates need to be regulated in order to limit the incidence of apnea immediately after birth at a physiologically tolerable level.

We have shown that increased Mafa protein levels in 4A mutant mice likely results in increased $Gad2$ transcription as indicated by our in vitro transcriptional analysis and the rate of apneas in vivo. $GAD_{65}$, synthetized from the $Gad2$ gene, is a rate-limiting enzyme responsible for GABA synthesis, especially during intense synaptic activities[48,49]. We propose that by mutating Mafa phosphorylation sites (4A mutation) and preventing Mafa phosphorylation by GSK3, $GAD_{65}$ levels are increased. Moreover, a similar apneic phenotype is seen when the mutation is expressed either in all Mafa+ neurons or when it was restricted to the inhibitory Mafa+ cell fraction. This indicated a prominent mediation of the effect through inhibitory GABAergic rather than excitatory Mafa neurons. This was confirmed by rescuing the apneic phenotype by systemic injection of a $GABA_A$ receptor antagonist. As increased presynaptic amount of cytosolic GABA, synthetized by $GAD_{65}$, is known to potentiate GABA release[50,51], it is likely that increased transcription of $Gad2$ results

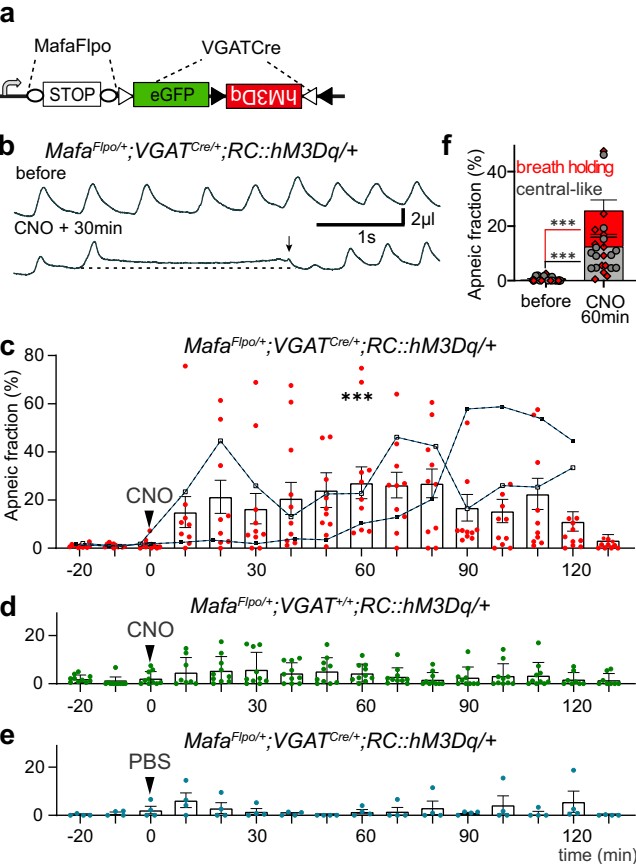

**Fig. 6 Chemogenic activation of Mafa inhibitory neurons triggers apneas.** **a** Genetic scheme to introduce the hM3Dq DREADD in Mafa+ inhibitory neurons. **b** Plethysmographic traces of the ventilation of a P1 $Mafa^{Flpo/+}$; $VGAT^{Cre/+}$; $RC$::$hM3Dq/+$ pups before (top) and 30 min after a subcutaneous injection of CNO (5 mg/kg). Note the presence of a click-like upward pressure shift terminating the breath holding apnea. **c** Injection of CNO in P1-2 mutants caused a five-fold increase of the apneic time fraction ($n = 12$ pups, Two-way ANOVA comparing the CNO effect on mutants and on $Cre^{+/+}$ siblings $P = 0.001$, df = 1, $F = 14.7$). This effect was usually reversible ($n = 10/12$ pups) but caused irreversible ($n = 2/12$ pups) apneic breathing (black traces filled and empty symbols) followed by death briefly after. **d, e** Experimental controls showing no effects of CNO on $Cre^{+/+}$ pups (**d**, $n = 10$ and no effect of saline (PBS) on mutant pups (**e**, $n = 4$). **f** Quantification of the apneic time fraction of apnea types before and 60 min after injection of CNO showing significant increases for both types ($n = 12$ pups; breath holding: two-sided Wilcoxon test $P = 0.0005$ and central-like: two-sided Wilcoxon test $P = 0.0005$). Plots represent mean ± sem.

in increased efficacy of GABAergic neurons harboring the 4A mutation. Indeed, Mafa phosphorylation by GSK3, through controlling cytosolic GABA level may impact synaptic vesicle filling and the size of the cycling vesicular pool, two interlocked processes through which augmented transmitter supply translates into augmented synaptic release[51]. Finally, it has been shown that GSK3 is de-activated in an activity-dependent manner through intracellular calcium-dependent activation of PI3K and Akt signaling pathways[52,53]. In Mafa+ inhibitory neurons, this activity-dependent inhibition of GSK3 activity would stabilize Mafa and in turn increased $Gad2$ transcription. Hence, our work reveals a mechanism whereby Mafa phosphorylation may ensure that synaptic GABA synthesis is adapted to activity. Therefore, our work demonstrates an unexpected regulatory role of GSK3 through Mafa on the weight of synaptic inhibition and provides a

novel means to displace the excitatory/inhibitory balance of activity through interference with discrete inhibitory neuronal subsets in model circuits. For example, hypo-activity of $Mafa^{4A}$ mutants suggests an impairment of locomotor circuits[54,55] that can now be addressed by intersectional genetic strategies, using the present $Mafa^{flox4A}$ and $Mafa^{Flpo}$ lines.

Mouse neonates presented with two distinct types of apneic events distinguishable by the lung inflation status and association to click sounds. About one half of apneic events in wild types were central-like apneas during which typically the lungs deflate normally, while the other half occurred while the lungs remain inflated, which is typical of "breath holds". Strikingly, $Mafa^{n4A/n4A}$ and $Mafa^{VGAT4A/VGAT4A}$ mutants showed an increase in the incidence of breath holding but not of central-like apneas, implying a central dysfunction other than in the rhythm generator which we confirmed was functional in vitro. The possibility that mutant breath holding apneas correspond to apneustic events (i.e., prolonged maintenance of inspiratory tone[56]) is unlikely. First, apneustic breathing requires deficient drives from both vagal afferents and pontine respiratory structures that would have disrupted the general pattern of breathing[57,58]. Second, breath holds deployed after inspiration had ceased and were at times accompanied by tentative inspiratory efforts, two features incompatible with an ongoing apneustic "inspiratory cramp". Third, apneusis cannot be induced pharmacologically in mouse neonates[59]. Finally, breath holding apneas causing partial or complete retention of the inspired air in the lung were selectively accompanied by a production of a click time-locked to sudden deflations of the lung terminating the breath holds. This temporal organization is reminiscent of the re-opening of closed airways observed during vocal breathing albeit in the absence of the powerful expiratory effort compressing the lung and enabling call emission (this study and[9]). Although clicks have unknown origin, in these two breathing contexts they could strikingly stand as acoustic signatures of the re-opening of closed airways.

The present observations suggest that in mutants, stabilization of Mafa in inhibitory premotor neurons controlling upper airway patency might favor obstructions. We show that this may apply to the geniohyoid muscle known to increase airway patency. In addition, we identify breath holds as major targets of Mafa-related controls. Although the obstructive nature of breath holds would require further investigation, it is strongly supported by the ability to maintain lung inflation, the resuming of rhythmic inflation efforts and the final elicitation of a click, otherwise known to correspond to the opening of airway cavities from a closed state, a concept that has phonological relevance, in association to USV emission in mice (this study and[9]) and as a regular part of the consonant systems of many human languages[60]. This issue is important clinically, because example cases of airflow limitation during exhalation have been described in children following sighs[61,62] and breath holds pushing air against closed airways are a common feature of abnormal breathing in Rett syndrome (RTT) patients[3,63,64]. In $Mafa^{n4A/n4A}$ mutant pups, the inspirations that precede breath holds were not augmented. In a mouse model of RTT, post-inspiratory breath holds where air is pushed against closed airways are thought to result from insufficient[43,65] GABAergic drives to the KF that would entail prolongation and augmentation of post-inspiratory upper airway adductive, and abdominal muscle expiratory, drives[43,66]. In $Mafa^{n4A/n4A}$ mutant pups, the onset of the obstructive event would take place after inspiration has ceased in agreement with an impairment of the post-inspiratory coordination of pharyngeal muscles contractions albeit in the absence of noticeable joint expiratory efforts. Furthermore, Mafa+ inhibitory neuronal targets are lacking in primary and secondary sensory vagal and trigeminal neurons (including the nTS, the putative source of

inhibitory drive to the KF thought to be depressed in RTT[43]). Breath holding apnea akin to Rett syndrome and breath holds associated to the $Mafa^{4A}$ mutation, although sharing post-inspiratory airway closure, appear to arise from orthogonal contexts regarding (i) GABAergic drives respectively deficient (Rett) and augmented ($Mafa^{4A}$) and (ii) the association to pronounced expiratory efforts, respectively present (Rett) and absent ($Mafa^{4A}$) and thus should not be confounded. As the $Mafa4A$ mutation likely spares respiratory reflexes and the excitability of pontine respiratory circuits the deficit may rather owe to enhanced inhibitory premotor drive onto upper airway opener muscles favoring their collapse.

How increased inhibitory drives from Mafa+ neurons come to set the condition and the location (e.g., palate, tongue base, lateral walls, epiglottis) of airway collapse is not known. Mafa+ inhibitory neurons are resident in the peri12 and peri-nAmb areas. In the peri12, a fraction of them are premotor neurons to the geniohyoid, a muscle whose activity during resting breathing remains controversial[67,68] but that activates during inspiratory-resistive breathing and during airway occlusion possibly through active eccentric contraction to regulate the position of the hyoid bone and increase upper airway size and stability[69]. Airway obstruction is unlikely caused by defective contraction of the sole geniohyoid muscle, further tracing experiments are required to check whether the same geniohyoid (through axonal collateralization[70]) or other Mafa+ inhibitory, premotor neurons lie upstream hypoglossal motoneurons innervating the genioglossus muscle, the other main pharyngeal opener muscle (reviewed in Ref. [15]), that also locate for part in the peri12 area[71,72]. Although the hypoglossal nucleus is often described as inspiratory, hypoglossal motoneurons were additionally categorized as pre-inspiratory, inspiratory/post-inspiratory and even expiratory discharging neurons while these patterns have been shown to evolve in context dependent manners[73–75]. This makes it possible that depressed hypoglossal motor drives impact upper airway patency outside of inspirations, notably and timely, during post-inspiration. Indeed, reduction of the hypoglossal motor drive during post-inspiration, a time when it should normally be reflexively increased by laryngeal inflation-related sensory drives[75] (likely spared by the 4A mutation), could precipitate airway collapse and cause limited expiratory flow.

Nothing is presently known of the Mafa+ inhibitory neurons in the peri-nAmb area, the possibility that they might be premotor to pharyngeal constrictor motoneurons that reside in the nucleus Ambiguus will need to be investigated. Paradoxically, when the airway volume is relatively small, inhibition of pharyngeal constrictor muscle tone further enhances pharyngeal wall flaccidity, and favors pharyngeal collapse[14]. The normal vocalizing behavior of Mafa mutants indicates preservation of adductive movements at laryngeal level. Thus, if obstructive, breath holding apneas of Mafa mutants likely originate in the pharynx mostly composed of soft tissues, thus most susceptible to collapse[76]. Such a "valve-like" breathing behavior affecting exhalation but not inhalation has been proposed to result from palatal prolapse in Obstructive Sleep Apnea (OSA) patients[77,78].

We here identified Mafa as a marker for a set of inhibitory neurons of the reticular formation with probable impact on upper airway patency at birth. Interestingly, we found that 60% of Mafa+ neurons in the peri-hypoglossal reticular formation but not those of the peri-nAmb that flank the nucleus ambiguus, are V1 type neurons with a history of expression of the homeobox gene $Engrailed1$ (Supplementary Fig. 6i, k). The spatial proximity with somatic Mo12 motoneurons and the triple V1, Mafa+ and inhibitory nature of these cells are evocative of spinal Renshaw cells[25] involved in recurrent inhibition of somatic spinal motoneurons[79,80]. Altogether, Mafa phosphorylation by GSK3,

by modulating GABAergic inhibitory transmission, may impact the resistance of upper airways through controlling the output gain of motor drives at both premotor and, via recurrent inhibitions, motor levels. To get further insights on the role of Mafa[+] inhibitory neurons in breathing circuits, it will be important, capitalizing on the present molecular signatures, to identify their presynaptic partners in pontine[81] and medullary sites thought to generate post-inspiratory drives[82] and in sleep related areas given the prevalence of obstructive sleep apneas[83].

The present study points to the possibility that abnormal *Mafa* gene function may contribute to human respiratory pathologies. It is noteworthy that Toruner et al.[84] identified duplication and translocation of chromosomal 8q24.3 region encompassing the *MAFA* locus in a few cases of SIDS. An increase in MAFA protein level comparable to the one we describe, triggered by lack of phosphorylation, could occur through increased synthesis in case of gene duplication or other genomic alterations. SIDS has many etiologies, and upper-airway obstruction may be one of them[85]. Pharyngeal collapse occurs in OSA without fatal issue, in 3–7% of adults and 1–4% of children. Lower and posterior position of the hyoid bone is widely recognized as a major trait in OSA patients[14], and the heritability of this trait has been demonstrated[14,86,87]. Backward position of the hyoid bone can result either from intrinsic craniofacial features or from hypotonic supra-hyoid muscles as suggested here. Potential involvement of *MAFA* in OSA was also suspected in a human genetic study on whole genome scan for OSA susceptibility loci[88]: indeed, the highest linkage with the apnea/hypopnea index was observed in the 8q24 region encompassing the *MAFA* locus. In addition, numerous epidemiologic studies point to comorbidity between OSA and diabetes mellitus for which the role of *MAFA* is well-established (reviewed in Ref. [89]). MAFA dysregulation may be the link between these two pathologies.

## Methods

**Housing**. Animals were group-housed with free access to food and water in controlled temperature conditions (room temperature 21–22 °C, humidity 40-50%), and exposed to a conventional 12-h light/dark cycle. All experimental procedures and the handling of mice were done in accordance with the European Community Directive 86/609/EEC and following the recommendations of the French National Ethics Committee for Science and Health report on "Ethical Principles for Animal Experimentation" - CEEA n°59 Paris Centre et Sud – under agreement N° 2015071710462096. All efforts were made to reduce animal suffering and minimize the number of animals.

**Mouse genetics**. To generate the Mafa[flox4A] allele, Mafa-4A cDNA was cloned by PCR and introduced into the targeting vector using SalI - NruI cloning sites containing a neomycine cassette flanked by two Flp sites for ES cell selection and later removal. A Diphteric Toxin (DTA) cassette was inserted on the 3' end of the construct (after the 2 kb 3' genomic region) to enhance homologous recombination rate. Genomic DNA from G418 selected ES cells clones were screened by Southern Blot: DNA was digested by ScaI then, a 1.8 kb 3' external probe hybridized with either the recombinant Mafa [flox4A] (8.8 kb), the recombinant Mafa [floxLacZ] (11 kb) or the wild type Mafa (6.2 kb). Positive clones were confirmed by digestion with BglII and hybridization with a 1 kb 5' external probe and an internal probe corresponding to the neomycine cassette. Integrity of the loxP site and ORF were further confirmed by PCR. From the 8 positive ES clones obtained, 4 were injected into blastocysts and 2 of them gave rise to germ line transmission. The *Mafa[flox4A]* mice were genotyped by the presence or absence of the 5' loxP site by PCR1: (forward primer 5'-AGCTAGGGGGAGAGAGGCCCGCG-3' and reverse primer 5'-GTGCTGAGGGGCGTCGAGGACAGCGA-3'), and the presence of the 4A mutation by PCR2: (forward primer 5'-GCCGCGGAGCTGGCGATGGG-3', reverse primer 5'-GCTGAGGGGGGCCGAGGACAGGGC-3', Supplementary Fig. 1). *Mafa[flox4A]* heterozygote intercrosses yielded offspring in Mendelian distribution (Supplementary Table 4).

To generate the *Mafa[Flpo]* allele, an optimized version of the Flpo recombination enzyme was inserted into the *Mafa* locus, together with a neomycin positive and DTA negative selection cassette. Southern blots performed using genomic DNA digested with PfIFI and hybridized with a 5' external probe, resulted in a 12.7 kb band for WT and 10.1 kb in the recombinant. Southern blots using genomic DNA digested with NheI hybridized with a 3' external probe resulted in a 10.7 kb fragment for WT and 6.7 kb after recombination. *Mafa[Flpo]* animals were genotyped

for the presence of Flpo by PCR (forward primer 5'-CAGCTTCGACATCGTG AACA-3', reverse primer 5'-ACAGGGTCTTGGTCTTGGTG-3', Supplementary Fig. 7). *Mafa[floxLacZ]* animals were genotyped for the presence of Mafa (forward primer 5'-GAGGCCTTCCGGGGTCAGAGCTTCG-3', reverse primer 5'-TG TTTCAGTCGGATGACCTCCTCCTTG-3') and for the presence of LacZ (forward primer 5'-CGGCGGAATTCCAGCTGAGCGCCGGTCT-3', reverse primer 5'-AG ACCGGGGCGGCCTGCGCAAACTT-3'). *Nestin[Cre/+], VGAT[Cre/+] and En1[Cre/+]* animals (Supplementary Table 5) were genotyped for the presence of Cre (forward primer 5'-GCGGTCTGGCAGTAAAAACTATC-3', reverse primer 5'-GTGAAA CAGCATTGCTGTCACTT-3') and an internal PCR control (forward primer 5'-CTAGGCCACAGAATTGAAAGATCT-3', reverse primer 5'-GTAGGTGGA AATTCTAGCATCATCC-3'). *RC::FeLa* mice (Supplementary Table 5) were genotyped for the presence LacZ (forward primer 5'-AGTTCACCCGTGCACC GC-3', reverse primer 5'-CGCTCGGGAAGACGTACG-3') and the presence of GFP (forward primer 5'-AAGACCCGCGCCGAGGTGAAGT-3', reverse primer 5'-CGCCGATGGGGGTGTTCTGC-3'). *RC::FL-hM3Dq* mice (Supplementary Table 5) to identify wild type and mutant alleles (respectively forward primer 5'-AAGGGAGCTGCAGTGGAGT-3', reverse primer 5'-CAGGACAACGCCCACAC A-3' and forward primer 5'-TGTATCCAGGAGGAGCTGATG-3', reverse primer 5'-GGAGCAACATAGTTAAGAATACCAG-3'). *Mafa[LacZ/+]* and *Mafa[4A/+]* mice were kept on 129 Sv background whereas *Mafa[floxLacZ/+], Mafa[flox4A/+]* and *Mafa[Flpo/+]* were back-crossed on C57Bl6/J background.

**Phenotypic analysis**. Experimental procedures and the handling of mice were done in accordance with European regulations following the recommendations of the local ethics committee CEEA59. All phenotypic analyses were performed blind on naturally born pups that were genotyped afterwards. The birth time (P0) of pups was determined (±0.5 h) and P0 pups were placed for 10 min under a heating lamp where they were gently touched until their breathing had stabilized. The weight, temperature of mouse neonates, the presence of suckling-like jaw movements induced by touching their lip with a foam tip and body movements (measured as percentage of time over 5 min observation periods) were examined. Vocalizations were recorded in five minutes continuous audio recording periods (see below).

**Plethysmographic and audio recordings, detection and analysis of respiratory cycles, apnea definition**. Breathing variables were measured in unanaesthetized, unrestrained animals by whole-body barometric plethysmography as described previously[90]. Three hundred second continuous records were acquired at defined time points after birth (P0 = less than 1 h after birth, between 1 and 2 h after birth (P0 + 2 h), 3–5 h after birth (P0 + 4 h), 7–9 h (P0 + 8 h) after birth, 11–13 (P0 + 12 h) hours after birth and at P1, P2 days after birth). The plethysmographic chamber (20 mL) was connected to a reference chamber of equal volume and the inter-chamber pressure difference was measured using a differential pressure transducer (Validyne DP-103-14) connected to a demodulator (Validyne CD15). The plethysmographic signals recorded in the absence of limb or body movements were sampled at 1 kHz, stored on a computer and analyzed using the Elphy2 software (developed by G Sadoc at CNRS). The durations of inspirations (Ti) and expirations (Te) were measured to calculate breath frequency $F_R$ ($F_R = 1/Ti + Te$; breath/min); tidal volume, $V_T$ (microL); and ventilation $V_E$ ($V_E = F_R x V_T/1000$ μL/g/min). Calibrations were performed by injecting 2.5 μL of air in the chamber with a Hamilton syringe. The $CO_2$ chemo-reflex was tested by measuring ventilation changes induced by a two minute exposure to 8% $CO_2$ enriched air.

The temporal pattern of breathing in each recorded sample was characterized as regular or apneic from Poincaré plot (Fig. 1b). A regular breathing pattern consisted in a population of repetitive cycles, the distribution of which could be fitted by a single Gaussian, and was analyzed as such, by measuring mean durations and standard deviations. An apneic breathing pattern was a mix of short repetitive cycles (regular breathing cycles followed and preceded by cycles of about the same duration) and of long isolated cycles (apneic cycles followed and preceded by short cycles). Apneic cycles were defined as being longer than a "cut-off respiratory cycles" (CORC) duration, measured in Poincaré plots from the longest repetitive cycles: CORC is located at the boundary between the regular breathing population (forming an ovoid cloud close to the origin) and the population of isolated apneic cycles (spreading along the x and y axis away from the diagonal, Fig. 1b).

Alternatively, we defined apneic respiratory cycles as cycles longer than a threshold duration (ThD). Using ThD's classically used to define apneas (e.g., ThD = 3 s or longer in mice), we found that the total duration of apneic cycles (apneic time fraction, ATF) is proportional to log(ThD). Robustness of this relationship (whatever the ThD is), suggests that the constant ATF/log(ThD) ratio is an homogeneous characteristic of the entire apneic population. Thus, we extrapolated ThD toward shorter cycles (e.g., ThD < 3 s), until a limit beyond which ATF/log(ThD) differs significantly from the typical apneic value. This limit measures the CORC; ATF for ThD = CORC gives the total time spent in apnea; the number of apneas is the number of cycles longer than CORC. In all recording samples, identical values were obtained using our alternative definition of apneas (i.e., isolated events), indicating that both methods depict the same population of apneic, isolated respiratory cycles.

Breath holding apneic events in plethysmographic recordings were characterized by (i) a post-inspiratory airway closure (filled arrowheads in Fig. 1e, f) causing partial

or complete retention of the inspired air in the lung (plateau above the basal end-expiratory volume) either followed by (ii) waning and resuming of rhythmic inspiratory air inflows of progressively larger tidal volumes (empty arrowheads, Fig. 1e) or (iii) the presence at the end of the breath holds of a brief (<1 ms) audio-mechanical event named "click" (Figs. 1f, 3e, 6b and Supplementary Fig. 3b). Click emissions were found temporally associated to inspiratory on-switch and termination of lung compression enabling USVs during vocal breathing sequences (Supplementary Fig. 3a). Note that the analysis of plethysmographic recordings during vocalizations (Supplementary Fig. 3a), must prominently take into account lung compression, thus the relationship between pressure, flow rate and resistance of upper airways[91] (Boyle law) in addition to the vaporization of water during inspiration (Ideal gas law) itself prominent during resting breathing. Similarly during breath holding apneic breathing clicks were time-locked both to inspiratory on-switch, and the termination of breath holding apneas (Fig. 1f and Supplementary Fig. 3b). Whether clicks associated to inspiratory on-switch also denote an end-expiratory obstructive context remains to be investigated. Audio recordings were obtained with an UltraSoundGate condenser microphone capsule CM16 (sensitive to frequencies from 20 Hz to 180 kHz) and Avisoft Recorder software (sampling rate, 250 kHz; format, 16 bit) from Avisoft Bioacoustics. To synchronize the acquisitions of plethysmographic and audio signals, a "send trigger" command is written into the Elphy program. This generates a 10 ms trigger pulse sent to both an analog input of the automated data collection system acquiring the plethysmographic signal and the Avisoft UltraSound Gate system acquiring the audio signal. Although synchronization of the two signals can be obtained within 1-ms accuracy, the clocks of the two systems have a ±50 ppm rating, which can result in deviations of ≥10 ms/min of recording. To take this drift into account, we gently knocked on the plethysmographic chamber with a metal rod before the end of the recording session to create synchronous signals of a pressure change and sound that were used after acquisition to adjust the timing for both signals. Repeated tests demonstrated that this procedure ensured that the latencies between the plethysmographic and the audio signals were <2 ms in a 10-min recording. Elphy configuration and program files are available at http://yzerlaut.github.io/Elphy/.

A total of 1070 clicks were counted in joint audio-plethysmographic recordings (5 min duration) from WT animals ($n = 6$) and found preferentially associated to vocal breathing (81.5 ± 4.6 %) and breath holding apneic breathing (15.0 ± 3.7%) while very scarce during eupneic and central apneic breathing (2.2 ± 0.7% and 1.1 ± 0.5% respectively, Supplementary Fig. 3e). Examination of the occurrence of clicks in relation to respiratory contexts was performed on 5 s samples during vocalizing events ($n = 70$ samples from 4 wild type pups), eupnea ($n = 40$ samples from 3 wild type pups) and breath holding events ($n = 15$ samples from 5 Mafa$^{n4A/n4A}$ mutant pups). This analysis was not performed during central apneic events for lack of clicks. In these samples, plethysmographic traces were aligned and centered on the inspiratory on-switch ($n = 47$ events) and end lung compression ($n = 34$ events) for vocal breathing (1 s time window, Supplementary Fig. 3a), on small amplitude upward pressure shifts ($n = 25$ events) for breath holds (2 s time window, Supplementary Fig. 3b) and aligned on the inspiratory on-switch and centered at approximately a mid-respiratory cycle time ($n = 40$ events) for eupneic breathing (1 s time window, Supplementary Fig. 3c). The about 50% value of the mode of the distribution (central 20 ms mode bin) for vocal breathing is due to the temporal excursion allowing click detections timed to the preceding or the following events. Central-like apneic events were characterized by a respiratory pause following a normal breath while the lung was deflated. Using these criteria breath holding vs central like apnea were validated manually and independently by two investigators, when the nature of the apneic cycle was found ambiguous (<1% of apneic cycles) the cycle was ruled as a central-like apneic cycle.

### Pentylene tetrazole (PTZ) injections and chemogenetic activation of Mafa+ inhibitory neurons

We performed PTZ dose response analysis (2, 10, 20, 40 and 80 μg/animal) in mouse neonates to determine the maximal PTZ dose (40 μg/animal) that did not induce an epileptic crisis. A single subcutaneous injection of 0.02 ml (40 μg) PTZ was performed at P0 + 12 h. Plethysmographic signals were acquired just before and 30 min after treatment in control and Mafa$^{n4A/n4A}$ mutants. A third group of Mafa$^{n4A/n4A}$ was injected with isotonic NaCl following the same procedure, to control specificity of the treatment. For chemogenetic experiments, VGAT$^{Cre/+}$; Mafa$^{Flpo/+}$ mice were mated with RC$^{FL-hM3Dq/FL-hM3Dq}$. In the resulting litter, Mafa$^{Flpo/+}$; RC$^{FL-hM3Dq/+}$ pups were first identified by GFP expression in the lens. Cre genotyping was performed afterwards. Plethysmography recordings (duration: 300 s) were performed at P1 every 10 min for 30 min before a single subcutaneous injection of CNO (clozapine N-oxide, ENZO lab, 5 mg/kg).

### Brainstem-spinal cord preparation

The methods used for preparing brainstem-spinal cord preparations from P0 mice and maintaining them in oxygenated artificial cerebrospinal fluid (aCSF) have been described. Briefly, brainstem spinal cord preparations were dissected in 4 °C aCSF of the following composition (in mM): 125 NaCl, 3 KCl, 1.2 CaCl$_2$, 1 MgCl$_2$, 25 NaHCO$_3$, 5 KH$_2$PO$_4$, 30 glucose, pH 7.4. The preparation was then transferred to the recording chamber (2 ml) on a net and superfused (2 ml/min) with oxygenated aCSF at 27 °C. Phrenic nerve roots were recorded using glass suction electrodes (150 μm tip diameter). The micropipettes filled with aCSF were connected through silver wires to a high-gain AC amplifier (7P511; Grass, West Warwick, RI), bandpass filtered (3 Hz to 3 kHz), recorded and integrated

(time constant, 100 ms) on a computer via a digitizing interface (CED micro 1401) and analyzed using spike2 software (CED, Cambridge, UK).

### Viral Tracing experiments and histology

Briefly, mouse pups between ages P2-P4 were anaesthetized by hypothermia and two microliters of a viral solution containing equal volumes of Rb-ΔG-mCherry (titer ~1e + 8) and a helper virus, HSV-hCMV-YTB (titer ~3e + 8) was pressure injected unilaterally into the geniohyoid muscle via a glass micro-pipette. Five days post-injection, the pups were transcardially perfused with cold PBS and 4% PFA and the brains collected for histological processing. Serial sections (30 μm thick, every 2nd section) were imaged either with a Leica TCS SP8 confocal microscope or using a slide scanner (Hamamatsu NanoZoomer S210).

Mafa expressing cells in Mafa$^{LacZ}$ embryos were visualized using a ßGal staining procedure. Briefly, dissected brainstem at different embryonic stages were fixed 2 h in 1% formaldehyde 0.2% glutaraldehyde in washing solution (2 mM MgCl$_2$, 0.01% sodium deoxycholate, 0.02% NP40, 5 mM EGTA, all in phosphate buffer saline) and stained for 2–4 h in a solution with 1 mg/ml X-Gal (5 mM potassium ferricyanide, K$_3$[Fe(CN)$_6$], 5 mM potassium ferrocyanide, K$_4$[Fe(CN)$_6$];3H$_2$O, 2 mM MgCl$_2$, 0.01% sodium deoxycholate, 0.02% NP-40, in PBS).

The following antibodies were used for immunostainings: rabbit anti-Mafa (Bethyl IHC-00352, 1/200); chicken anti-ßGal (Abcam ab9361, 1/2000); goat anti-ChAT (Millipore AB 144P, 1/1000), rabbit anti-RFP (Rockland 600-401-379, 1/500), chicken anti-GFP (Aves, GFP1020, 1/1000), rabbit anti-NK1R (Sigma, S8305, 1/5000). The primary antibodies were revealed by the following secondary antibodies of the appropriate specificity: donkey anti-rabbit Cy5 (Jackson laboratories, 712-165-153), donkey anti-chicken 488 (Jackson laboratories, 703-545-155), donkey anti-chicken Cy5 (Jackson laboratories, 703-176155), goat anti-rabbit Cy3 (Invitrogen,A10520), donkey anti-goat Cy5 (Jackson laboratories, 705-606-147), all used at 1:500 dilution.

Endogenous expression of Mafa protein was detected, after demasking in citrate bufffer 10 mM pH6, using rabbit anti-Mafa polyclonal antibody (Bethyl IHC-00352; dilution 1/200 in PBS 0,05% Tween 20) followed either with a secondary goat anti-rabbit Cy3 antibody (Invitrogen, A10520). Since Mafa protein was barely detectable in wild type animals at P0 we also used a double amplification system: first with avidin/biotin (ABC kit, Vector) coupled to peroxidase, and then Tyramide System Amplification (TSA Plus, Perkin Elmer, coupled to Cy3).

In situ hybridization was performed on 14 μm thick frozen section. cDNA clones encoding mouse Gad1 (IMGSp981H028), mouse Gad2 (IMAGp998N0514452), mouse Glyt2 (IRCKp5014K088Q) were purchased from Source Bioscience, mouse VGlut2 (440 bp) was amplified with the following primers 5'-GGCCACCGGAT CCTCCCCTT-3' and 5'-ATAGCGGAGCCTTCTTCTCAG-3' and cloned into PCR Blunt vector. Digoxigenin-labeled RNA probes were synthesized from linearized cDNA templates using dig RNA labeling kit (Roche 11175025910). Frozen sections were hybridized overnight at 65 °C in hybridization buffer (50% formamide, 1,3X SSC, 5 mM EDTA, 50 μg/ml yeast RNA, 0,2% tween20, 0.5% CHAPS, 100 μg/ml heparin) rinsed at 65 °C in 1 × SSC 50% formamide. The transcripts were detected using anti-digoxigenin antibody coupled to alkaline phosphatase (Roche 11093274910) with NBT and BCIP substrates.

### Fluorescent in situ hybridization

Hindbrain, vagal and trigeminal ganglia were obtained from P0 mice pups (C57Bl6 background) fixed in 4% paraformaldehyde. Multiplex single molecule Fluorescent In Situ Hybridization (smFISH) was performed on 14 μm thick cryosections using a RNA-scope v2 Multiplex kit (Advanced Cell Diagnostic) following manufacturer instructions. The following probes were used Mafa (C1), Phox2b (C2) and VGlut2 (C3) and revealed using the respective Tyramide substrate (Cy3, Opal520 and C5). In one set of experiment, the Phox2b probe was replace by a mix Gad1/Gad2 probe design in C2 channel. All probes were obtained from Advanced Cell Diagnostic company. Section were counter labeled using DAPI and slide mounted using ProLong Diamond antifade medium. Images were acquired using a confocal Zeiss LSM700, processed using Image J and Photoshop.

### Cell culture conditions and granule cell progenitors' purification

Medulloblastoma cell lines DAOY and ONS-76 were cultured in MEM and RPMI 1640 medium (GIBCO), respectively, supplemented with 10% fetal bovine serum (GIBCO), 100 units/mL penicillin, and 100 μg/mL streptomycin (Invitrogen), 1.25 μg/ml fungizone (Invitrogen). DAOY Medium was supplemented with 0.1 mM non-essential amino acids and 1 mM sodium pyruvate. Granule cell progenitors (GCP) were purified from mouse cerebella dissected at P7. Following dissociation (30 min, 37 °C) in trypsin/DNAse solution at 37 °C and trituration in DNAse, cells were separated on a density step gradient of 35% and 60% Percoll solution (Sigma). Purified GCPs were further enriched by panning on tissue culture dishes to remove adherent fibroblasts. Non adherent cells were plated at a density of $4 \times 10^6$ cells/mL in 12 wells plate coated with a poly D-lysine solution (Sigma) and Matrigel (BD Biosciences), with 12.5 μg/mL SHH (R&D). Cells were grown in Neurobasal medium with B27 supplement, 2 mM glutamine, and 100 U/mL penicillin/streptomycin (all from Invitrogen), linoleic acid-albumin, 0.45% D-glucose, and 16 μg/mL N-acetyl-cysteine, 1% SPITE (all from Sigma-Aldrich). After

1 h plating, GCPs were infected, washed and cultured (37 °C, 5% $CO_2$) in their medium for 48 h when their RNA was extracted.

**Real time RT-qPCR, constructs, transfection and retroviral production**. Total RNAs were extracted using RNeasy Plus mini kit (Qiagen) and reversely transcribed using Cloned AMV First-Strand cDNA Synthesis Kit, Invitrogen). Quantitative real-time PCR assays (see primer list on Supplementary Table 6) were conducted using SYBR Green real-time PCR Master Mix and real-time PCR amplification equipment (Applied Biosystem). The RT-qPCR analysis was normalized using *TBP* expression.

The pcDNA3-*Mafa* or *Mafa-4A* plasmids were obtained by inserting the *Mafa* or *Mafa-4A* coding region into the BamH1 and EcoRI restriction sites of the pcDNA3 vector (Invitrogen). Medulloblastoma cells DAOY and ONS-76 were transfected using Effectene reagent (Qiagen). Transfected cells were selected with neomycin. The retroviral constructs pMIGR-*Mafa* or *Mafa-4A* were obtained by inserting the *Mafa* or *Mafa-4A* coding region into pMIGR. Retroviruses were produced in 293T cells, by co-transfecting retroviral pMIGR-derived vectors and the packaging plasmids pCAG-4 and pMD.gag-pol, using lipofectamine 2000 (Invitrogen). Retroviral particles were harvested 48–72 h after transfection.

**Statistical analysis**. All data are reported as mean ± sem. Normal distributions of data points were tested using D'Agostino & Pearson or Shapiro–Wilk tests. One- and Two-way ANOVA was used to compare groups of data, unpaired and paired Student's *t*-test were used to compare two Gaussian distributions of continuous variables, non-parametric tests (Mann–Whitney or Wilcoxon) were used otherwise. Chi square ($\chi^2$) test was used for discrete variables. All graphs and statistical analyses were generated using GraphPad Prism software. $*p < 0.05$, $**p < 0.01$, $***p < 0.001$.

**Statistics and reproducibility**. For all experiments, the number of experiments is indicated in the legends of the relevant figure. Histological analyses and tracing experiments were reproduced a minimum of three times.

**Reporting summary**. Further information on research design is available in the Nature Research Reporting Summary linked to this article.

## Data availability
The data that support the findings of this study can be found in the Source Data provided with the paper. Microscopy data are available from the corresponding authors upon reasonable request.

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

## Acknowledgements

We deeply thank Noura Mebirouk, Christophe Alberti, Frederic Bertrand, Francoise Ruelle, Jessy Leloup, Magalie Larcher, Valerie Lavallée, Krystel Saroul and Sandra Autran for transgenic animals production and maintenance. C.P. is Equipe labellisée Ligue Contre le Cancer. This work was supported by ANR-15-CE16-0013-02 and ANR-19-CE16-0029-02 grants to G.F.

## Author contributions

Conceptualization, L.L., C.P., A.E. and J.C.; Methodology, L.L., N.R., C.P., S.B. and M.G.; Investigation, L.L., M.-P. M.-S., B.D., A.G., P.-L. R. and C.P.; Supervision, C.P., J.C. and G.F.; Project administration, A.E. and G.F.; Funding acquisition, A.E., C.P., J.C. and G.F.; Writing, G.F with contributions from all authors.

## Competing interests

The authors declare no competing interests.

## Additional information

**Correspondence** and requests for materials should be addressed to . Present address: Institut de Biologie de l'École Normale Supérieure IBENS, École Normale Supérieure, CNRS, INSERM, PSL Université Paris, 75005 Paris, FranceLaure Lecoin, Celio Pouponnot or Gilles Fortin.

