## [Peer Review File · Nature Communications]

Mafa-dependent GABAergic activity promotes mouse neonatal apneasReviewers' comments:

Reviewer #1 (Remarks to the Author):

The authors show that the transcription factor Mafa is non-essential for survival and yet a specific mutation of this protein (Mafa4A) causes massive postnatal lethality. The authors attribute the animals' rapid demise to their inability to breathe normally and they identify the specific breathing anomaly elicited by this mutation as apneas of the breath-holding type possibly caused by airway obstruction. By restricting the Mafa4A mutation to neurons, the authors demonstrate that the lethality and the breathing disorder are of neural origin. They show that these apneas are not caused by a central chemoreceptor anomaly because the hypercapnic chemoreflex is preserved and the retrotrapezoid nucleus which is responsible for a large portion of this homeostatic reflex does not express Mafa. In addition, the authors show that the core of the breathing pattern generator (core circuit responsible for breathing automaticity) is minimally affected given that the inspiratory neural outflow observed in isolated brainstems maintained in vitro is normal. These observations lead the authors to the logical conclusion that the defect could be related to airway control (some form of obstructive apnea). Using a mutant created for this purpose (MafafloxLacZ26) they show that Mafa is heterogeneously expressed in the brainstem and is principally expressed by inhibitory (gad2+) neurons. They also provide evidence that Mafa4A mutation enhances gad2 expression. Evidence that the apneas result from excessive GABA release rests on two converging lines of evidence. First the incidence of apneas is reduced by acute administration of a subconvulsive dose of pentylenetetrazole, a noncompetitive antagonist of the gamma-aminobutyric acid (GABA)(A) receptor complex. Second, selective activation of Mafa-expressing inhibitory neurons (VGAT^{cre/+};Mafa^{Flopo/+} mice crossed with R26FL-hM3Dq/FL-hM3Dq to express a Gq-coupled DREADD exclusively in neurons positive for both VGAT and Mafa) reproduces the breathing disorder observed in Mafa4A/n4A mutants. Finally, the authors demonstrate, using a retrograde transsynaptic vector, that Mafa is expressed by a population of inhibitory premotor neurons that control the genio-hyoid muscle suggesting that these particular Mafa-expressing inhibitory neurons could be responsible for the postulated airway obstructions.

This work is outstanding, conceptually and technically. The control experiments and statistical treatment are appropriate. This team of investigators are top experts in brainstem development and breathing physiology. Their methodology (transgenic mice, vectors, techniques to record breathing in neonate mice, histology etc.) is state of the art. The discussion is scholarly and more than appropriate but the authors may want to consider the following issues.

1. How do the authors know that the apneas are the cause of death? Are the animals dramatically hypoxic/hypercapnic?
2. Could the authors have dismissed a little too fast the possibility that the Mafa4A mutation causes a breath-holding type of apnea akin to Rett syndrome rather than airflow obstruction? They did not provide incontrovertible evidence that these apneas were obstructive. The evidence is based on plethysmography which only measures airflow. The authors did show that some inhibitory airway premotor neurons express Mafa and they also showed that activating Mafa+ inhibitory neurons at large mimics the effect of the Mafa4A mutation. However, they did not specifically show that the airway premotor neurons were responsible for the apneas.
3. Couldn't the "breath-hold-like" apneas be caused by the intrusion of thoracic and abdominal

muscle contractions unrelated to breathing (postural)?

4. The “breath-hold” apneas of the Rett pathology have been tentatively attributed to an issue with GABA transmission in the dorsolateral pons (Kölliker-Fuse nucleus specifically), a region considered essential for airway control and certainly replete with inhibitory neurons (<https://pubmed.ncbi.nlm.nih.gov/26507912/>). Neonatal isolated brainstem-spinal cord preparations do not provide much information on what happens in the parabrachial region. The authors do not mention specifically whether Mafa is expressed in this brain region. Does Figure 2b include the KF/lat parabrachial region?

5. Line 193: The authors appear to dismiss the possibility that the sensory trigeminal nuclear complex, where Mafa is abundantly expressed, could contribute to a defect in airway control. This may be a little premature. Upper airway (e.g. laryngeal) sensory afferents innervate the paratrigeminal nucleus (Pr5) via the jugular ganglion (cf. work of AE McGovern and colleagues) and this pathway contributes to upper airway reflexes. Are the authors certain that Mafa is not expressed in the Pr5?

Reviewer #2 (Remarks to the Author):

Lecoin et al. provides a thorough characterization of how the Mafa protein impacts respiration. They conduct phenotypic analysis of breathing using both loss and gain of function alleles and discover that gain of function Mafa mutants are lethal and have a considerable number of apneas as neonates. Then, they provide a potential molecular mechanism for the apnea by describing a group of brainstem neurons that perhaps have increased GABAergic signaling to motor neurons for an upper airway muscle. In general, the experiments are rigorously conducted, the multiple transgenes and genetic approaches are comprehensive and sophisticated, and the statistics are complete. While the arch of the story is plausible, two aspects should be re-vised or experimentally re-visited: 1) the demonstration that these apneas are actually due to upper airway obstruction, and 2) that the entirety of the apnea and lethality can be attributed to the small group of geniohyoid inhibitory premotor neurons.

1) Definition of obstructive versus central apnea. Within the medical literature, obstructive apneas are defined as absent nasal airflow despite persistent respiratory effort (abdominal or diaphragm activity). As the authors point it, this is due to upper airway collapse (Jordan et al. Lancet 2014). It is not obvious that this type of apnea can only occur during expiration, and in fact, a hallmark of obstructive apneas is snoring which involves collapse of the upper airway during inspiration. In contrast, central apneas are when both nasal airflow and respiratory effort is absent. To rigorously define these two types of apneas, the authors should either measure inspiratory muscle activity or abdominal movement during the apneas. This is particularly important for several reasons:

- a. the manuscript is framed as being specific to obstructive apneas
- b. the geniohyoid is innervated by the hypoglossal nucleus, which is inspiratory, so it is unclear how increased inhibition during expiration would lead to obstruction (Kubin Comparative Physiol 2016).
- c. Results presented in Figure 3h and Figure 6 do not show specificity for obstructive apneas.

2) What mice are being used for controls in each experiment? In figure 1g, it is apparent that the

genetic makeup of the mouse is important for the basal apneic time fraction. The control mice have a reduced apnea time fraction compared to WT mice. In the legend it states that the control mice are littermates, however, the control mice used for the comparison to Mafa(n4a/n4a) in Figure 1f/g are the same used for Figure 3h when compared to Vgat4a mice. The proper littermates for these experiments should be used as controls. Additionally, since Mafa(flox4a/flox4a) is partially lethal (only 60% survive), are the littermate controls Mafa(flox4a/flox4a) without Nestin or Vgat Cre?

3) Proportion of mice that survive to weaning. Vgat-cre mice are only 55% lethal as compared to 100% lethality of Mafa4a and nearly 100% for Mafa-n4a. This lethality nearly matches that of the Mafa(flox4a/flox4a) line. Since Vgat-Mafa4a mice have similar apneic fraction time to the nestin-cre line, this suggests that the lethality cannot be completely explained by the apnea.

4) The model proposed by PTZ and increased Gad2 expression is that geniohyoid motor neurons receive increased gabaergic inhibition. However, this is never directly measured. Increased gabaergic neurotransmission could be measured geniohyoid motor neurons in Vgat-cre-Maf4a by electrophysiology. If motor neurons are labeled, then IPSP's during preBötC bursts or tonic IPSP's could be measured.

5) Is there a reason why the N used in Figure 1g is different than 1c?

Reviewer #3 (Remarks to the Author):

The study entitled “Mafa-dependent GABAergic activity promotes neonatal obstructive apneas” by Dr. Laure Lecoin and colleagues is an interesting study examining the role of Mafa in the control of breathing and neonatal mortality.

The major finding of this study is that Mafan4A/n4A mutant mouse pups had a regular breathing pattern in the first hour post birth, but breathing rapidly deteriorated in mutants through an abnormal increase in the incidence of respiratory pauses. The authors should be commended for such a thorough and multi-dimensional series of experiments. However, while the data is clear, the interpretation is not.

The authors suggest this respiratory dysfunction is mainly caused by changes in the central control of airway resistance, but a more likely interpretation of the data presented is that peripheral sensory pulmonary mechanosensitive neurons are most affected.

This hypothesis also is more consistent with other work in the field that indicates mouse embryos deficient in c-Maf display abnormal development of sensory neurons, particularly low-threshold mechanoreceptive sensory neurons (PMID: 22514301).

The typical plethysmography recordings shown in figure 1 have a downward drift of baseline – why is this occurring? While not mandatory – most work in the field presents inspiration as downward deflections.

The inspiratory effort seen in the plethysmography recordings (figure 1) seems prolonged – but not restrictive. There is no evidence of obstructive or flow limited inspiratory events in these examples.

As the authors indicate, the abnormal breathing consisted more of an “abrupt post inspiratory reduction of expiratory flow that prolonged lung inflation”.

Taken together this indicates it is unlikely there is a problem with increased upper airway resistance, as the authors propose, but more likely a dysfunction of inspiratory termination. This could be due to inadequate feedback from pulmonary stretch receptors, in keeping with the hypothesis that peripheral sensory mechanosensitive neurons are most affected in the Mafan4A/n4A mutant mouse pups.

Recording tongue EMG activity would directly address this major concern. Is tongue EMG activity during inspiration from mutant mice less when compared to tongue EMG activity from WT mice?

The brain localizations shown in figure 2 are not convincing, at least for implicating hypoglossal motorneurons and airway resistance, as the authors suggest. Most of the density is in zones that could be considered more nucleus tractus solitarius (NTS) – a nuclei that contains sensory afferent pulmonary synapses. In the ventral brainstem there is a high density in the preBotzinger and Botzinger complexes – regions thought to be essential for respiratory rhythmogenesis and post-inspiratory activity.

An additional important question for proper interpretation of the data, both mechanistically, and for relevant to SIDS, include - were there changes in the ECG in mutant mouse pups? And, if so, did ECG abnormalities (arrhythmia, bradycardia) precede or follow, alterations in respiratory activity?

The double labeling of Mafa+/mCherry+ with virally labeled neurons (Figure 4) lateral to hypoglossal motorneurons is interesting, but it is premature to call them “genio-hyoid premotor neurons” due to their location. The lateral population of GABA neurons shown in figure 4 may, or may not, synapse upon hypoglossal motor neurons.

Overall, this study contains a considerable amount of very interesting data for the field.

However, the manuscript needs significant re-interpretation of the data, and additional experiments focused on the inclusion and analysis of the ECG and tongue EMG activity would provide much needed additional insights to the mechanistic and disease-related aspects of this mutation.

Response to reviewers

Overview

We are grateful for the positive assessment of our work, and the constructive suggestions for improvement. In response, we have performed a number of additional experiments to address the major points – specifically: 1) to further assess the obstructive nature of breath holding apneas in Mafa4A mutants; and 2) demonstrate that altered breathing of Mafa4A mutants pups is unlikely caused by deficit of the sensory arm of respiratory control.

In preview, this entailed new functional and anatomical experiments in which we show through joint plethysmographic and audio recordings that brief (ms) broad-band audio-mechanical events called “clicks” that we had previously found time-locked to re-opening of close airways to generate calls during vocal breathing are also selectively time locked to breath hold expiratory terminations. We also demonstrate using RNAscope and an intersectional genetic background that Mafa+ inhibitory neurons are absent in primary vagal and trigeminal sensory ganglia and in their secondary sensory target Pa5, Pr5 and nTS. We have addressed all other queries (but one for loss of the line) and our original conclusions are upheld.

Our comments to queries are in blue, the relevant text changes in the ms in red, modified or new figures and a new supplementary table are appended at the end of this document (Figures 1,2,4; Supplementary Figure 3,6; Supplementary Table 4).

Reviewer #1 (Remarks to the Author):

The authors show that the transcription factor Mafa is non-essential for survival and yet a specific mutation of this protein (Mafa4A) causes massive postnatal lethality. The authors attribute the animals' rapid demise to their inability to breathe normally and they identify the specific breathing anomaly elicited by this mutation as apneas of the breath-holding type possibly caused by airway obstruction. By restricting the Mafa4A mutation to neurons, the authors demonstrate that the lethality and the breathing disorder are of neural origin. They show that these apneas are not caused by a central chemoreceptor anomaly because the hypercapnic chemoreflex is preserved and the retrotrapezoid nucleus which is responsible for a large portion of this homeostatic reflex does not express Mafa. In addition, the authors show that the core of the breathing pattern generator (core circuit responsible for breathing automaticity) is minimally affected given that the inspiratory neural outflow observed in isolated brainstems maintained in vitro is normal. These observations lead the authors to the logical conclusion that the defect could be related to airway control (some form of obstructive apnea). Using a mutant created for this purpose (MafafloxLacZ26) they show that Mafa is heterogeneously expressed in the brainstem and is principally expressed by inhibitory (gad2+) neurons. They also provide evidence that Mafa4A mutation enhances gad2 expression. Evidence that the apneas result from excessive GABA release rests on two converging lines of evidence. First the incidence of apneas is reduced by acute administration of a subconvulsive dose of pentylenetetrazole, a noncompetitive antagonist of the gamma-aminobutyric acid (GABA)(A) receptor complex. Second, selective activation of Mafa-expressing inhibitory neurons (VGATcre/+;MafaFlpo/+ mice crossed with R26FL-hM3Dq/FL-hM3Dq to express a Gq-coupled DREADD exclusively in neurons positive for both VGAT and Mafa) reproduces the breathing disorder observed in Mafa4A/n4A mutants. Finally, the authors demonstrate, using a retrograde transsynaptic vector, that Mafa is expressed by a population of inhibitory premotor neurons that control the genio-hyoid muscle suggesting that these particular Mafa-expressing inhibitory neurons could be responsible for the postulated airway obstructions.

This work is outstanding, conceptually and technically. The control experiments and statistical

treatment are appropriate. This team of investigators are top experts in brainstem development and breathing physiology. Their methodology (transgenic mice, vectors, techniques to record breathing in neonate mice, histology etc.) is state of the art. The discussion is scholarly and more than appropriate but the authors may want to consider the following issues.

We are grateful for these positive comments on our work. We have now explored in greater detail the breathing disturbances of Mafa mutants. In short, we complement our earlier data by showing that breath holding apneas (i) arise from an obstructive upper airway context and (ii) differ from the type of breath holding apneas akin to Rett syndrome.

1. How do the authors know that the apneas are the cause of death? Are the animals dramatically hypoxic/hypercapnic?

Analyzing the cause of death is beyond the scope of most reports that deal with a lethal phenotype. This said, our work characterizes in mouse pups a clear apneic mutant phenotype (decreased minute ventilation, constant V_T) that develops and worsens over the first day after birth with parallel appearance of manifest cyanotic profiles and increased morbidity. Mutant pups although hypo-active were able to vocalize, suckle and had normal weights ruling out dietary deficits. We thus considered apneas as the most logical cause of lethality. Further support to this view came from experimental series showing that (i) none (0/10) of PTZ treated pups at P0+12h, when the incidence of apnea is maximal, died during the period when PTZ was active while half (5/10) of them died in the next two hours after apneic breathing had resumed, (ii) 2/12 pups in DREADD experiments developed irreversible apneic breathing after CNO injection and died in the two hours following the time when 10/12 mutants had recovered baseline breathing activity. Altogether, this strongly suggested that apneas are the proximal cause of death of the mutants. Given that we have not measured blood saturations we have chosen a less assertive formulation concerning causality and removed “leading to” in our introductory sentence:

Text change:

Line 101: “Mafa-4A mutants at birth showed abnormally high incidences of obstructive apneas and died within 48 hours.”

2. Could the authors have dismissed a little too fast the possibility that the Mafa4A mutation causes a breath-holding type of apnea akin to Rett syndrome rather than airflow obstruction?

We thank the reviewer for this suggestion. We have now revised the manuscript considering the Rett syndrome issue through both production and discussion of new data. We have now (to also respond to reviewer 3), looked at the presence of Mafa+ inhibitory neurons in cranial sensory ganglia (trigeminal and vagal), in their secondary sensory target territories the Pa5, the Pr5 and the nTS and in pontine respiratory areas bearing in mind that impaired sensory control there may be causal to breath holding apneas akin to Rett Syndrome (details are appended to the response to your point 4) .

They did not provide incontrovertible evidence that these apneas were obstructive. The evidence is based on plethysmography which only measures airflow.

Performing electromyographic recording of inspiratory and/or abdominal muscles in unrestrained apneic neonates (also a concern of reviewer 2) raised limiting both technical and ethical issues. Alternatively, we now complement plethysmographic recordings with joint audio recordings of clicks previously found time-locked to re-opening of closed airways following lung compressions that precede innate neonatal utterances. Examining further the temporal organization of clicks in breathing

contexts, we found that clicks were virtually absent during eupneic and central-like apneic breathing but were, in addition to vocal breathing, selectively associated with Mafa mutant breath-holding apneic breathing. More precisely, clicks were time-locked to abrupt expiratory-like lung deflations terminating breath holds. We propose that these findings further support the view that the *Mafa4A* mutation predisposes to obstructive breathing through promoting upper airway collapsibility.

We now present these data in a new panel on **Fig. 1f** and on a new **Supplementary Figure 3** (see end of document).

Corresponding text changes are in Results, Discussion and Material and Methods

Text change (red) in Results:

Line 178: “In breath holding apneas, the slow deflation of the lung could be accompanied by a resuming of rhythmic inflation efforts that progressively, on a cycle-to-cycle basis, showed increased tidal volumes and likely reflected the progressive re-opening of the airways (open arrowheads in Fig.1e) or could be terminated by abrupt expiratory-like lung deflations (Fig. 1f). To gain indirect insights on the status of airways during breath holds, we monitored audio correlates of upper airway function (see Materials and Methods), more precisely, we focused on brief (ms order) broad-band audio-mechanical events named “clicks” or cracking sounds (see refs 9,30,31) that correspond to openings of airway cavities from a closed state^{32,33}. Indeed, clicks have been reported to often associate with ultrasonic vocalizations (USVs) ^{30,34} and, more precisely, were found temporally locked to lung post-compressions’ re-openings of closed airways powering calls during vocal breathing⁹ (Supplementary Fig.3a). We found that breath holding apneas, like vocalizations, were accompanied by clicks (Fig.1f) systematically time locked to small amplitude upward deflections of the plethysmographic trace (downward arrows in Figs 1f, 3e, 6b and Supplementary Fig.3b) that immediately precede expiratory-like lung deflations terminating breath holds. Interestingly, clicks were virtually absent during both (non-vocal) eupneic breathing (Supplementary Fig.3c) and central-like apneas (Supplementary Fig.3d,e). Altogether, these data strongly support the obstructive nature of breath holding apneas in mutant pups.”

Text changes (red) in Discussion

Line 422: “Finally, breath holding apneas causing partial or complete retention of the inspired air in the lung were selectively accompanied by a production of a click time-locked to sudden deflations of the lung terminating the breath holds. This temporal organization is reminiscent of the re-opening of closed airways observed during vocal breathing albeit in the absence of the powerful expiratory effort compressing the lung and enabling call emission (this study and ⁹). Although clicks have unknown origin, in these two breathing contexts they strikingly stand as acoustic signatures of the re-opening of closed airways.”

Text changes (red) in Material and methods

Line 624: “Breath holding apneic events in plethysmographic recordings were characterized by (i) a post-inspiratory airway closure (filled arrowheads in **Fig.1e,f**) causing partial or complete retention of the inspired air in the lung (plateau above the basal end-expiratory volume) either followed by (ii) waning and resuming of rhythmic inspiratory air inflows of progressively larger tidal volumes (empty arrowheads, Fig.1e) or (iii) the presence at the end of the breath holds of a brief (<1 ms) audio-mechanical event named “click”. Click emissions were found temporally associated to inspiratory on-switch and termination of lung compression enabling USV production during vocal breathing

sequences (**Supplementary Figure 3a**). Similarly during breath holding apneic breathing clicks were time-locked both to inspiratory on-switch, and the termination of breath holding apneas (**Figure 1f and Supplementary Fig. 3b**). Whether clicks associated to inspiratory on-switch also denote an end-expiratory obstructive context remains to be investigated. Audio recordings were obtained with an UltraSoundGate condenser microphone capsule CM16 (sensitive to frequencies from 20 Hz to 180 kHz) and Avisoft Recorder software (sampling rate, 250 kHz; format, 16 bit) from Avisoft Bioacoustics. To synchronize the acquisitions of plethysmographic and audio signals, a “send trigger” command is written into the Elphy program. This generates a 10ms trigger pulse sent to both an analog input of the automated data collection system acquiring the plethysmographic signal and the Avisoft UltraSound Gate system acquiring the audio signal. Although synchronization of the two signals can be obtained within 1-ms accuracy, the clocks of the two systems have a ± 50 ppm rating, which can result in deviations of ≥ 10 ms/min of recording. To take this drift into account, we gently knocked on the plethysmographic chamber with a metal rod before the end of the recording session to create synchronous signals of a pressure change and sound that were used after acquisition to adjust the timing for both signals. Repeated tests demonstrated that this procedure ensured that the latencies between the plethysmographic and the audio signals were < 2 ms in a 10-min recording. The relevant Elphy configuration (.gfc) and program (.pg2) files are available at neuro-psi.cnrs.fr/IMG/ElphyConfigVocal/breath7-usb.gfc and neuro-psi.cnrs.fr/IMG/ElphyConfigVocal/breath7-usb.pg2). A total of 1070 clicks were counted in joint audio-plethysmographic recordings (5 min duration) from WT animals (n=6) and found preferentially associated to vocal breathing (81.5 ± 4.6 %) and breath holding apneic breathing (15.0 ± 3.7 %) while very scarce during eupneic and central apneic breathing (2.2 ± 0.7 % and 1.1 ± 0.5 % respectively, **Supplementary Fig 3e**). Examination of the occurrence of clicks in relation to respiratory contexts was performed on 5 second samples during vocalizing events (n=70 samples from 4 wild type pups), eupnea (n=40 samples from 3 wild type pups) and breath holding events (n=15 samples from 5 *Mafa*^{n4A/n4A} mutant pups). This analysis was not performed during central apneic events for lack of clicks. In these samples, plethysmographic traces were aligned and centered on the inspiratory on-switch (n=47 events) and end lung compression (n=34 events) for vocal breathing (1s time window, **Supplementary Fig 3a**), on small amplitude upward deflections (n=25 events) for breath holds (2s time window, **Supplementary Fig 3b**) and aligned on the inspiratory on-switch and centered at approximately a mid-respiratory cycle time (n= 40 events) for eupneic breathing (1s time window, **Supplementary Fig 3c**). The about 50% value of the mode of the distribution (central 20ms mode bin) for vocal breathing is due to the temporal excursion allowing click detections timed to the preceding or the following events.”

The authors did show that some inhibitory airway premotor neurons express Mafa and they also showed that activating Mafa+ inhibitory neurons at large mimics the effect of the Mafa4A mutation. However, they did not specifically show that the airway premotor neurons were responsible for the apneas.

We agree with the reviewer that proof of necessity would be nice. In a strict sense, our claim would be best supported by transsynaptic ChR2 labelling experiments, however, these experiments are exceedingly difficult, if not impossible, because the rabies viruses would kill cells before they express enough opsin for successful optogenetic interference. In addition, this experiment is *not assured* since there are likely several other premotor/motor/muscle pools contributing to set the resistive status of

upper airways (as now discussed for ex. premotor neurons to the genioglossus that locate in the peri12 area, see text change in the Discussion below).

Text change in the Discussion:

Line 455: “. Airway obstruction is unlikely caused by defective contraction of the sole geniohyoid muscle, further tracing experiments are required to check whether the same geniohyoid (through axonal collateralization⁷⁰) or other Mafa⁺ inhibitory, premotor neurons lie upstream hypoglossal motoneurons innervating the genioglossus muscle, the other main pharyngeal opener muscle (reviewed in¹⁵), that also locate for part in the peri12 area^{71,72}.”

3. Couldn't the “breath-hold-like” apneas be caused by the intrusion of thoracic and abdominal muscle contractions unrelated to breathing (postural)?

We thank the reviewer for this suggestion, we surmise that the referee means a Valsalva maneuver-like behavior which would correspond to an expiratory effort against closed airways. Our data indeed support a closed status of the airways during breath-holds however, direct visual or video monitoring of mutant pups attested to their overt hypo-active state and apart from seldom and sudden jerks or twitches failed to reveal any noticeable postural changes or chest wall movements suggesting active expiratory efforts during breath-holds. We have added this in the results.

Text change in Results:

Line 122: “Mutants neonates were hypoactive and **apart from seldom jerks or twitches, did not show postural nor chest wall movements suggesting exacerbated inspiratory or expiratory efforts,** developed cyanosis and died within 12 hours (**Supplementary Fig.1e,f**).”

4. The “breath-hold” apneas of the Rett pathology have been tentatively attributed to an issue with GABA transmission in the dorsolateral pons (Kölliker-Fuse nucleus specifically), a region considered essential for airway control and certainly replete with inhibitory neurons (<https://pubmed.ncbi.nlm.nih.gov/26507912/>). Neonatal isolated brainstem-spinal cord preparations do not provide much information on what happens in the parabrachial region. The authors do not mention specifically whether Mafa is expressed in this brain region. Does Figure 2b include the KF/lat parabrachial region? (we surmise the reviewer meant to say replete with inhibitory “neuronal terminals” rather than “neurons”, see ref. 44).

Per the Reviewer's suggestion, we have now provided better illustrations showing that Mafa⁺ inhibitory neurons are not resident in the PB/KF and have addressed the concern about sensory control of pontine respiratory groups experimentally by both examining the presence of Mafa⁺ inhibitory sensory neurons in vagal ganglia and in their projection territory the nTS, the putative main source of inhibitory drive to the KF thought to be depressed in Rett syndrome (see also our response to reviewer 3). Our data revealed the virtual absence of Mafa⁺ inhibitory neurons at these levels ruling out candidate neuronal targets in these structures.

We have now produced more convincing anatomical panels presented sequentially in a new **Fig.2** focalizing on absent Mafa⁺ neurons at the level of the respiratory rhythm generator (preBötC and RTN) and their presence in peri-hypoglossal and peri-nucleus ambiguus areas where their inhibitory nature is revealed; in a new **Fig. 4** where we use an intersectional genetic background to identify Mafa⁺/VGAT⁺ (Mafa⁺ inhibitory) neurons that are shown to be absent at pontine level in the PB,KF,Pr5 and at medulla level in the Pa5 and nTS.

Although likely sharing post-inspiratory closure of upper airways, breath-holding apneas akin to Rett syndrome and the present breath holding apneas arise from orthogonal (i) status of GABAergic transmission, respectively deficient and augmented, (ii) association with pronounced expiratory efforts, respectively present and absent and thus should not be confounded.

This is now discussed and referenced:

Text change in the results:

Line 264: “We refined the search for candidate Mafa⁺ inhibitory neurons considering that breath holding apneas may result from (i) altered sensory control of breathing originating in trigeminal and vagally-derived sensory neurons innervating the larynx and the lungs that mediate powerful protective reflexes that slow down or pause breathing^{41,42} and (ii) altered GABAergic transmission in dorsolateral pontine respiratory areas as the case in a mouse model of the Rett syndrome⁴³. Using RNAscope we report that the jugular and nodose vagal ganglia are by and large devoid of Mafa⁺ neurons (**Fig .4a**) and, likewise in the trigeminal sensory ganglia that hosts Mafa⁺ cells (**Fig .4b**), when present these neurons co-expressed VGlut2 thus were excitatory (**Fig .4c**). Next, using the intersectional line *VGAT^{Cre/+};Mafa^{floxLacZ/+}* (*Mafa^{VGATLacZ}*), inhibitory Mafa⁺ neurons were also found absent from the secondary sensory structures targeted by the above sensory afferents, the paratrigeminal nucleus (Pa5), the principal trigeminal nucleus (Pr5) and the nucleus of the solitary tract (nTS). Furthermore, pontine dorsal respiratory areas comprising the Kölliker-Fuse and the parabrachial nucleus also lacked Mafa⁺ inhibitory neurons (**Fig.4d**). We conclude that Mafa mutant apneic breathing is unlikely caused by alterations of respiratory sensory feedbacks or their integration at medullary of pontine levels.”

Text change in the discussion:

Line 429: “Example cases of airflow limitation during exhalation have been described in children following sighs^{61,62} and breath holds pushing air against closed airways are a common feature of abnormal breathing in Rett syndrome (RTT) patients^{3,63,64}. In *Mafa^{n4A/n4A}* mutant pups, the inspirations that precede breath holds were not augmented. In a mouse model of RTT, post-inspiratory breath holds where air is pushed against closed airways are thought to result from insufficient^{44,65} GABAergic drives to the KF that would entail prolongation and augmentation of post-inspiratory upper airway adductive, and abdominal muscle expiratory, drives^{44,66}. In *Mafa^{n4A/n4A}* mutant pups, the onset of the obstructive event took place after inspiration had ceased in agreement with an impairment of the post-inspiratory coordination of pharyngeal muscles contractions albeit in the absence of noticeable joint expiratory efforts. Furthermore, Mafa⁺ inhibitory neuronal targets are lacking in primary and secondary sensory vagal and trigeminal neurons (including the nTS, the putative source of inhibitory drive to the KF thought to be depressed in RTT⁴⁴). Breath holding apnea akin to Rett syndrome and breath holds associated to the *Mafa^{4A}* mutation, although sharing post-inspiratory airway closure, appear to arise from orthogonal contexts regarding (i) GABAergic drives respectively deficient (Rett) and augmented (*Mafa^{4A}*) and (ii) the association to pronounced expiratory efforts, respectively present (Rett) and absent (*Mafa^{4A}*) and thus should not be confounded. As the *Mafa4A* mutation likely spares respiratory reflexes and the excitability of pontine respiratory circuits the deficit may rather owe to enhanced inhibitory premotor drive onto upper airway opener muscles favoring their collapse.”

5. The authors appear to dismiss the possibility that the sensory trigeminal nuclear complex, where Mafa is abundantly expressed, could contribute to a defect in airway control. This may be a little premature. Upper airway (e.g. laryngeal) sensory afferents innervate the paratrigeminal nucleus (Pr5)

via the jugular ganglion (cf. work of AE McGovern and colleagues) and this pathway contributes to upper airway reflexes. Are the authors certain that Mafa is not expressed in the Pr5?

As a follow-up to your previous point, the same investigations were made concerning the trigeminal ganglion and its projection territories in the Pa5 and Pr5 with the same conclusion about the overall absence of Mafa+ inhibitory neurons therein.

This point has been addressed above in response to your point 4. We now make reference to the work of AE McGovern and colleagues (our ref. 42)

Reviewer #2 (Remarks to the Author):

Lecoin et al. provides a thorough characterization of how the Mafa protein impacts respiration. They conduct phenotypic analysis of breathing using both loss and gain of function alleles and discover that gain of function Mafa mutants are lethal and have a considerable number of apneas as neonates. Then, they provide a potential molecular mechanism for the apnea by describing a group of brainstem neurons that perhaps have increased GABAergic signaling to motor neurons for an upper airway muscle. In general, the experiments are rigorously conducted, the multiple transgenes and genetic approaches are comprehensive and sophisticated, and the statistics are complete. While the arch of the story is plausible, two aspects should be re-vised or experimentally re-visited: 1) the demonstration that these apneas are actually due to upper airway obstruction, and 2) that the entirety of the apnea and lethality can be attributed to the small group of geniohyoid inhibitory premotor neurons.

We are grateful for these positive comments on our work.

*1) Definition of obstructive versus central apnea. Within the medical literature, obstructive apneas are defined as absent nasal airflow despite persistent respiratory effort (abdominal or diaphragm activity). As the authors point it, this is due to upper airway collapse (Jordan et al. Lancet 2014). It is not obvious that this type of apnea can only occur during expiration, and in fact, a hallmark of obstructive apneas is snoring which involves collapse of the upper airway during inspiration. In contrast, central apneas are when both nasal airflow and respiratory effort is absent. To rigorously define these two types of apneas, the authors should either measure inspiratory muscle activity or abdominal movement during the apneas. This is particularly important for several reasons:
a. the manuscript is framed as being specific to obstructive apneas*

Performing electromyographic recording of inspiratory and/or abdominal muscles in unrestrained apneic neonates raised limiting both technical and ethical issues. Alternatively, we have now complemented plethysmographic recordings with joint audio recordings of clicks previously found time-locked to re-opening of closed airways following lung compressions that precede innate neonatal vocal emissions. Examining further the temporal organization of clicks in breathing contexts, we found that clicks were virtually absent during eupneic and central-like apneic breathing but were, in addition to vocal breathing, selectively associated with breath-holding apneic breathing. There, clicks were found time-locked to expiratory-like lung deflations terminating breath holds. We consider that these additional findings further support the obstructive nature of breath holding apneas in mutant pups.

We now present these data in a new panel **Fig. 1f** and on a new **Supplementary Figure 3**.

Text change in the results:

Line 178: “In breath holding apneas, the slow deflation of the lung could be accompanied by a resuming of rhythmic inflation efforts that progressively, on a cycle-to-cycle basis, showed increased tidal volumes and likely reflected the progressive re-opening of the airways (open arrowheads in Fig.1e) or could be terminated by abrupt expiratory-like lung deflations (Fig. 1f). To gain indirect insights on the status of airways during breath holds, we monitored audio correlates of upper airway function (see Materials and Methods), more precisely, we focused on brief (ms order) broad-band audio-mechanical events named “clicks” or cracking sounds (see refs 9,30,31) that correspond to openings of airway cavities from a closed state^{32,33}. Indeed, clicks have been reported to often associate with ultrasonic vocalizations (USVs) ^{30,34} and, more precisely, were found temporally locked to lung post-compressions’ re-openings of closed airways powering calls during vocal breathing⁹ (Supplementary Fig.3a). We found that breath holding apneas, like vocalizations, were accompanied by clicks (Fig.1f) systematically time locked to small amplitude upward deflections of the plethysmographic trace (downward arrows in Figs 1f, 3e, 6b and Supplementary Fig.3b) that immediately precede expiratory-like lung deflations terminating breath holds. Interestingly, clicks were virtually absent during both (non-vocal) eupneic breathing (Supplementary Fig.3c) and central-like apneas (Supplementary Fig.3d,e). Altogether, these data strongly support the obstructive nature of breath holding apneas in mutant pups.”

b. the geniohyoid is innervated by the hypoglossal nucleus, which is inspiratory, so it is unclear how increased inhibition during expiration would lead to obstruction (Kubin Comparative Physiol 2016).

We deem consider discussing this point further. Indeed, the hypoglossal nucleus inspiratory moto is certainly well comforted by the wealth of in vitro/ex vivo experimentation, but many in vivo evidence exist showing that the hypoglossal nucleus contains functionally heterogeneous subsets of motoneurons (including non-respiratory modulated ones) in line with the multiple functional implications of the tongue. Earlier work in cats reported a richer palette of firing discharges among hypoglossal motoneurons including pre-inspiratory/inspiratory, inspiratory/post-inspiratory and expiratory discharging ones (the latter more seldom) . A number of reports also showed the reconfiguration of hypoglossal discharges in diverse behavioral contexts (coughing, swallowing, emesis) and in relation to upper airway patency, both anticipatory pre-inspiratory and reflexive increased post-inspiratory discharges in response to laryngeal inflation have been reported. Our data indicates that breath holds are caused by post-inspiratory closure of the airways. We suspect that a reduced motor response to reflexive laryngeal sensory drives (that we now show likely spared by the 4A mutation see our response to reviewer 3) may favor post-inhibitory upper airway collapse. Although this scenario would be compatible with our data, the demonstration the Mafa+ inhibitory premotor neurons are indeed part of the laryngeal reflexive circuit will need to be investigated directly.

Text change in the discussion:

Line 460: “Although the hypoglossal nucleus is often described as inspiratory, hypoglossal motoneurons were additionally categorized *in vivo* as pre-inspiratory, inspiratory/post-inspiratory and even expiratory discharging neurons while these patterns have been shown to evolve in context dependent manners⁷²⁻⁷⁴. This makes it possible that depressed hypoglossal motor drives impact upper airway patency outside of inspirations, notably and timely, during post-inspiration. Indeed, reduction of the hypoglossal motor drive during post-inspiration, a time when it should normally be reflexively increased by laryngeal inflation-related sensory drives⁷⁴ (likely spared by the 4A mutation), could precipitate airway collapse and cause limited expiratory flow.”

c. Results presented in Figure 3h and Figure 6 do not show specificity for obstructive apneas.

We take it that this comment built on the somewhat unconvincing demonstration of the obstructive status of breath holding apneas. Towards this we consider that the additional audio recordings data sets (see our response to your point 1) strengthen the view that breath holding apneas have indeed an obstructive nature.

About **Figure 3h** (also taking into account your next point 2), we have corrected the datasets and confirmed the original results (in fact, in the corrected plot, the increased apneic time fraction owing to breath holding apneas in VGAT4A mutants compared to their flox4A/flox4A controls is slightly more significant (corrected: $P=0.0004$, $df=13.51$, $t=4.719$ vs previous $P=0.0018$, $df=31$, $t=3.41$), the corresponding corrected comparisons for central like apneas remain not significantly different ($p=0.5774$, $df=20$; $t=0.5663$ vs previous $p=0.1207$; $df=31$; $t=1.596$).

About **Figure 6**, we agree that breath holding and central apneas are both augmented in DREADD experiments and now provide an explanation for why this is the case using this intersectional genetic background. It still holds that CNO exposure increased the incidence of breath holding apneas, the fact that it impacts also central-like apneas is due to a spurious lineal heritage of DREADD by preBötC neurons that present with a transient embryonic phase of Mafa expression shown in a new **Supplementary Fig.6** described and explained in the results part.

Text change in the results:

Line 339: “We first verified that Mafa⁺ inhibitory neurons could be targeted by dual recombination with a Mafa^{Flopo} and a VGAT^{Cre} acting on the Rosa26^{FeLa} reporter line⁴⁷ (VGAT^{Cre/+};Mafa^{Flopo/+};RC::Fela/+), which specifically labels Mafa⁺/VGAT⁻ (excitatory) neurons with GFP and Mafa⁺/VGAT⁺ (inhibitory) neurons with bgal (**Supplementary Fig 6a-d**). We confirmed that Mafa⁺ neurons in the caudal medulla are almost exclusively inhibitory. In addition to the cell groups detected by ISH for Mafa in wildtype animals or by bgal immunostain in Mafa^{Lacz/+} mutants, this background surprisingly revealed doubly recombined neurons in the area of the preBötC respiratory rhythm generator, shown at birth, both devoid of Mafa expression (**Fig. 2; Supplementary Fig.6e**) and functional in Mafa^{n4A/n4A} mutants (**Supplementary Fig.4**). This is a likely consequence of an early and transient phase of Mafa expression in the rhombencephalon (**Supplementary Fig.6f,g**). With this in mind, we used the dual Mafa/VGAT recombinogenic background to trigger expression of a conditional G-protein coupled receptor hM3Dq-mCherry fusion DREADD (Designer Receptor Exclusively Activated by a Designer Drug⁴⁸) in all these neurons (**Fig. 6a, Supplementary Fig.6h**).”

...

Line 363: “CNO treatment increased the incidence of breath holding apneas in keeping with preceding results following knock-ins of the 4A mutation and PTZ pharmacological treatment. However, CNO treatment also augmented the incidence of central-like apneas likely through the spurious lineal access to preBötC neurons revealed above. We conclude that acute activation of Mafa⁺ inhibitory neurons *in vivo* is able to trigger breath holding apneas thus mimicking the phenotype of Mafa^{n4A/n4A} mice at birth.”

2) What mice are being used for controls in each experiment? In figure 1g, it is apparent that the genetic makeup of the mouse is important for the basal apneic time fraction. The control mice have a reduced apnea time fraction compared to WT mice. In the legend it states that the control mice are littermates, however, the control mice used for the comparison to Mafa(n4a/n4a) in Figure 1f/g are the same used for Figure 3h when compared to Vgat4a mice. The proper littermates for these experiments should be used as controls. Additionally, since Mafa(flox4a/flox4a) is partially lethal (only 60% survive), are the littermate controls Mafa(flox4a/flox4a) without Nestin or Vgat Cre?

We thank the reviewer for pointing out this discrepancy, in the original submitted manuscript for **Fig 1g**, and **Fig3g,h** we had erroneously pooled all Mafa flox4A animals for controls irrespective their

littermate status in crosses with nestinCre or VGATCre. We now only consider littermates for comparisons and have now corrected our data sets. This correction does not change the conclusion of these experiments.

In the new **Fig.1h** panel (former **1g**), we now compare *nestin cre/+ Mafa flox4A/flox4A (ncre/+*, n=10) mutants to their *nestin +/+ Mafa flox4A/4A (n+/, n=10)* control littermates. (unpaired t-test, P=0,016; df=9, f=2,368). In the **Fig.3g** and **h** panels, we now compare *VGATcre/+ Mafa flox4A/flox4A* mutants (*VGATcre/+*, n=10) to *VGAT+/+ Mafaflox4A/flox4A* controls (*VGAT+/+*, n=13).

Please see below our comment about the partial lethality of the *Mafa(flox4A/flox4A)* line.

3) Proportion of mice that survive to weaning. Vgat-cre mice are only 55% lethal as compared to 100% lethality of Mafa4a and nearly 100% for Mafa-n4a. This lethality nearly matches that of the Mafa(flox4a/flox4a) line. Since Vgat-Mafa4a mice have similar apneic fraction time to the nestin-cre line, this suggests that the lethality cannot be completely explained by the apnea.

The lethality of *Mafaflox4A/flox4A* in table 3 may be fortuitously misleading, the Kaplan-Meier plot in Supplementary **Figure 2a** shows that only one control (*nestin-Cre+/+;Mafaflox4A/flox4A*) pup out of 23 (4%) died on P1 in the P0-P5 time window. We have now also added an additional **Supplementary Table 4** called in Material and Methods – Mouse genetics- showing that intercrossing *Mafa^{flox4A}* heterozygotes yield a Mendelian distribution of offspring.

The survival disparity between nestin- and VGAT-cre mice presenting with equivalent apneic time fractions, convincingly incriminates inhibitory *Mafa* neurons in the apneic phenotype but as rightly pointed by this reviewer, indeed begs for additional causal factors to explain mortality. No simple trait (same hypoactive status and absence of dietary problems) could be associated to difference in survival outcomes. The possibility remains that equivalent apneic time fractions (admitted a coarse filter) may correspond upon closer examination to distinct temporal organization of apneas (frequency, durations, motifs) having distinct invasive character. Such an analysis was not performed, we have instead changed the text to explicitly mention this point and replaced the erroneous conclusive statement that assumed comparable morbidity of two conditional mutants.

Text change in the results:

Line 256: “Moreover, plethysmographic recordings indicated that *Mafa^{VGAT4A}* mutants showed an apneic time fraction double that of control littermates (**Fig.3e-g**) **accounted by a selective increase in the incidence of breath holding apneas (Fig.3h).** **The reason why *Mafa^{VGAT4A}* mutants show reduced morbidity compared to *Mafa^{n4A/n4A}* mutant is presently unknown.** These data show that restricting the 4A mutation to inhibitory neurons was sufficient to recapitulate the severe increase in the incidence of obstructive apneic events. ~~and explain morbidity.”~~

4) The model proposed by PTZ and increased Gad2 expression is that geniohyoid motor neurons receive increased gabaergic inhibition. However, this is never directly measured. Increased gabaergic neurotransmission could be measured geniohyoid motor neurons in Vgat-cre-Maf4a by electrophysiology. If motor neurons are labeled, then IPSP's during preBötC bursts or tonic IPSP's could be measured.

We fully agree with the reviewer. Unfortunately, we have lost the floxed4A line in the sequentially traumatic pandemic and lab moving contexts and re-establishment of the line is only about to start now after two unsuccessful revival rounds. This said, our work reveals tight links between *Mafa* phosphorylation, apneic breathing and impaired GABAergic transmission at birth based on

pharmacological and intersectional chemogenetic approaches, to our eyes we met the principal mechanistic linking demand by revealing the biochemical regulatory action of Mafa on GAD2.

5) Is there a reason why the N used in Figure 1g is different than 1c?

Yes, the scoring of breath holding vs central-like apnea was performed on a distinct set of P0+12h animals prior to PTZ injection. Also correcting for proper littermate controls (your point 2) the present **figure 1h** (former **figure 1g**) shows the data comparing ncre/+ (n=10) vs n+/+, (n=10).

Reviewer #3 (Remarks to the Author):

The study entitled “Mafa-dependent GABAergic activity promotes neonatal obstructive apneas” by Dr. Laure Lecoin and colleagues is an interesting study examining the role of Mafa in the control of breathing and neonatal mortality.

The major finding of this study is that Mafa^{n4A/n4A} mutant mouse pups had a regular breathing pattern in the first hour post birth, but breathing rapidly deteriorated in mutants through an abnormal increase in the incidence of respiratory pauses. The authors should be commended for such a thorough and multi-dimensional series of experiments. However, while the data is clear, the interpretation is not. The authors suggest this respiratory dysfunction is mainly caused by changes in the central control of airway resistance, but a more likely interpretation of the data presented is that peripheral sensory pulmonary mechanosensitive neurons are most affected.

We thank this reviewer for positive comments on our manuscript. The bulk of the present criticisms rests on the idea that we may have overlooked the putative contribution of impaired sensory control to explain the apneic phenotype of Mafa mutants whose obstructive status was also questioned. We thank the reviewer for raising these concerns.

In short, we complement our earlier data by showing that (i) candidate targets of the Mafa4A mutation (i.e. inhibitory Mafa⁺ neurons) are absent in trigeminal and vagal ganglia as well as in their projection territories the Pa5,Pr5 and NTS and (ii) breath holding apneas indeed arise in an obstructive upper airway context.

This hypothesis also is more consistent with other work in the field that indicates mouse embryos deficient in c-Maf display abnormal development of sensory neurons, particularly low-threshold mechanoreceptive sensory neurons (PMID: 22514301).

We respectfully would like to indicate that we manipulate Mafa not c-Maf and that low threshold mechanoreceptive sensory neurons were found grossly normal in *Mafa null* mutants (see Bourane et al., 2009 cited as ref. 26 in our ms), where their number and projections are maintained possibly via redundant effect of c-Maf. The invalidation of c-Maf whose expression at E11 precedes that of Mafa at E13.5 is often lethal at prenatal stages (leaving physiological deficits unexplored) while the conditional loss of function of c-Maf using *islet1cre* (to restrict the mutation to sensory neurons) indeed leads to severe alteration of the function of rapidly adapting mechanoreceptors yet these conditional mutants are healthy and fertile (see Wende et al. cited as ref. 29 in our ms). Finally, we demonstrate that Mafa inhibitory neurons are candidate targets of the Mafa4A mutation causing the deficit while such neurons are absent in vagal ganglia where candidate lung mechanoreceptors reside.

The typical plethysmography recordings shown in figure 1 have a downward drift of baseline – why is this occurring?

Our plethysmograph includes a constant leak ensuring its function as a “leaky” integrator. We illustrate raw traces not high pass filtered traces canceling DC drifts.

While not mandatory – most work in the field presents inspiration as downward deflections. The inspiratory effort seen in the plethysmography recordings (figure 1) seems prolonged – but not restrictive. There is no evidence of obstructive or flow limited inspiratory events in these examples. As the authors indicate, the abnormal breathing consisted more of an “abrupt post inspiratory reduction of expiratory flow that prolonged lung inflation”.

We agree with the reviewer on these points.

Taken together this indicates it is unlikely there is a problem with increased upper airway resistance, as the authors propose, but more likely a dysfunction of inspiratory termination. This could be due to inadequate feedback from pulmonary stretch receptors, in keeping with the hypothesis that peripheral sensory mechanosensitive neurons are most affected in the Mafan4A/n4A mutant mouse pups. Recording tongue EMG activity would directly address this major concern. Is tongue EMG activity during inspiration from mutant mice less when compared to tongue EMG activity from WT mice?

We respectfully consider the referee’s hypothesis as valid as our hypothesis that the abrupt post-inspiratory reduction of expiratory flow, during breath holds, results from a post-inspiratory closure of the airways involving deficiency of the motor rather than the sensory arm of respiratory control. Our revised manuscript now provides complementary functional data further supporting the obstructive nature of mutant breath holding apneas and anatomical data ruling out impact of the mutation on the sensory arm of respiratory control.

New functional data:

Performing electromyographic recording of inspiratory and/or abdominal muscles in unrestrained apneic neonates raised limiting both technical and ethical issues. Alternatively, we have now complemented plethysmographic recordings with joint audio recordings of clicks previously found time-locked to re-opening of closed airways following lung compressions that precede innate neonatal vocal emissions. Examining further the temporal organization of clicks in breathing contexts, we found that clicks were virtually absent during eupneic and central-like apneic breathing but were, in addition to vocal breathing, selectively associated with breath-holding apneic breathing. There, clicks were found time-locked to expiratory-like lung deflations terminating breath holds. We consider that these additional findings further support the obstructive nature of breath holding apneas in mutant pups.

We now present these data in a new panel (**Fig. 1f**) and on a new **Supplementary Figure 3**.

Text change in the results:

Line 178: “In breath holding apneas, the slow deflation of the lung could be accompanied by a resuming of rhythmic inflation efforts that progressively, on a cycle-to-cycle basis, showed increased tidal volumes and likely reflected the progressive re-opening of the airways (open arrowheads in Fig.1e) **or could be terminated by abrupt expiratory-like lung deflations (Fig. 1f). To gain indirect insights on the status of airways during breath holds, we monitored audio correlates of upper airway function (see Materials and Methods), more precisely, we focused on brief (ms order) broad-band audio-mechanical events named “clicks” or cracking sounds (see refs 9,30,31) that correspond to openings of airway cavities from a closed state^{32,33}. Indeed, clicks have been reported to often associate with ultrasonic vocalizations (USVs) ^{30,34} and, more precisely, were found temporally locked to lung post-compressions’ re-openings of closed airways powering calls during vocal breathing⁹ (Supplementary Fig.3a). We found that breath holding apneas, like vocalizations, were accompanied by clicks (Fig.1f)**

systematically time locked to small amplitude upward deflections of the plethysmographic trace (downward arrows in Figs 1f, 3e, 6b and Supplementary Fig.3b) that immediately precede expiratory-like lung deflations terminating breath holds. Interestingly, clicks were virtually absent during both (non-vocal) eupneic breathing (Supplementary Fig.3c) and central-like apneas (Supplementary Fig.3d,e). Altogether, these data strongly support the obstructive nature of breath holding apneas in mutant pups.”

New anatomical data:

Per the Reviewers’s suggestion (and also that of reviewer #1, see our response to his point 4), we have now provided better illustrations showing that Mafa+ inhibitory neurons are not resident in the PB/KF and have addressed the concern about sensory control of pontine respiratory groups experimentally by both examining the presence of Mafa+ inhibitory sensory neurons in vagal and trigeminal ganglia and in their projection territories the nTS, the Pr5 and Pa5. Our data revealed the virtual absence of Mafa+ inhibitory neurons at all these levels ruling out the presence of neuronal targets for the mutation in these structures and thus alterations of respiratory sensory feedbacks or their integration at medullary of pontine levels as causal to the present breathing alterations. These data are shown on a new **Figure 4** (see also our response to your next point below).

Text change in the results:

Line 264: “We refined the search for candidate Mafa+ inhibitory neurons considering that breath holding apneas may result from (i) altered sensory control of breathing originating in trigeminal and vagally-derived sensory neurons innervating the larynx and the lungs that mediate powerful protective reflexes that slow down or pause breathing^{41,42} and (ii) altered GABAergic transmission in dorsolateral pontine respiratory areas as the case in a mouse model of the Rett syndrome⁴³. Using RNAscope we report that the jugular and nodose vagal ganglia are by and large devoid of Mafa+ neurons (**Fig .4a**) and, likewise the trigeminal sensory ganglia that hosts Mafa+ cells (**Fig .4b**), when present these neurons co-expressed VGlut2 thus were excitatory (**Fig .4c**). Next, using the intersectional line $VGAT^{Cre/+};Mafa^{floxLacZ/+}$ ($Mafa^{VGATLacZ}$), inhibitory Mafa+ neurons were also found absent from the secondary sensory structures targeted by the above sensory afferents, the paratrigeminal nucleus (Pa5), the principal trigeminal nucleus (Pr5) and in the nucleus of the solitary tract (nTS). Furthermore, pontine dorsal respiratory areas comprising the Kölliker-Fuse and the parabrachial nucleus also lacked Mafa+ inhibitory neurons (**Fig.4d**). We conclude that Mafa mutant apneic breathing is unlikely caused by alterations of respiratory sensory feedbacks or their integration at medullary of pontine levels.”

Text change in the discussion:

Line 438: “Furthermore, Mafa+ inhibitory neuronal targets are lacking in primary and secondary sensory vagal and trigeminal neurons (including the nTS, the putative source of inhibitory drive to the KF thought to be depressed in RTT 44). Breath holding apnea akin to Rett syndrome and breath holds associated to the Mafa4A mutation, although sharing post-inspiratory airway closure, appear to arise from orthogonal contexts regarding (i) GABAergic drives respectively deficient (Rett) and augmented (Mafa4A) and (ii) the association to pronounced expiratory efforts, respectively present (Rett) and absent (Mafa4A) and thus should not be confounded. As the Mafa4A mutation likely spares respiratory reflexes and the excitability of pontine respiratory circuits the deficit may rather owe to enhanced inhibitory premotor drive onto upper airway opener muscles favoring their collapse.”

The brain localizations shown in figure 2 are not convincing, at least for implicating hypoglossal motorneurons and airway resistance, as the authors suggest.

Most of the density is in zones that could be considered more nucleus tractus solitarius (NTS) – a nuclei that contains sensory afferent pulmonary synapses. In the ventral brainstem there is a high density in the preBotzinger and Botzinger complexes – regions thought to be essential for respiratory rhythmogenesis and post-inspiratory activity.

Per the Reviewers's suggestion (and also that of reviewer 1, see our response to his point 4), we have now produced more convincing anatomical panels now presented sequentially in a new **Fig.2** focalizing on absent Mafa⁺ neurons at the level of the respiratory rhythm generator (preBötC and RTN) and their presence in peri-hypoglossal and peri-nucleus ambiguus areas where their inhibitory nature is revealed; in a new panel **Fig. 4d** where we use an intersectional genetic background to identify Mafa⁺/VGAT⁺ (Mafa⁺ inhibitory) neurons that are shown to be absent at pontine level in the PB,KF,Pr5 and at medulla level in the Pa5 and nTS.

Text changes have been mentioned in the response to your previous point above.

An additional important question for proper interpretation of the data, both mechanistically, and for relevant to SIDS, include - were there changes in the ECG in mutant mouse pups? And, if so, did ECG abnormalities (arrhythmia, bradycardia) precede or follow, alterations in respiratory activity?

We respectfully appreciate the interest for these questions, however this was not examined. Mechanistic interpretation here has the objective to link Mafa phosphorylation status to altered breathing at birth, adding cardiovascular examination is extravagantly ambitious, and falls outside the scope of this paper.

The double labeling of Mafa+/mCherry+ with virally labeled neurons (Figure 4) lateral to hypoglossal motoneurons is interesting, but it is premature to call them "genio-hyoid premotor neurons" due to their location. The lateral population of GABA neurons shown in figure 4 may, or may not, synapse upon hypoglossal motor neurons.

We should like to apologize for omitting a minimal methodological precision concerning the viral tracing method in the results indicating that the viral spread of deficient rabies viruses is **trans-synaptic** (our ms ref. 45) and constitutes the gold standard method to identify premotor neurons (see Stepien and Arber, Neuron 2010 or our own work Wu et al. Nat Comm 2017).

[In short, we injected in the geniohyoid muscle a cocktail of two neurotropic viruses. The first one is a genetically modified rabies viral vector to exploit its ability to spread through synapses from neuron to neuron (synaptic jump) in a retrograde manner (from the post-synaptic to the pre-synaptic neuron, i.e. from seed motoneurons to their premotor partners). These rabies viral vectors are deficient (G-deficient Rabies), a gene encoding a fluorescent protein (tracer) replaces a viral gene encoding the glycoprotein (G) critical for trans-synaptic spread. As such, if injected in the muscle, G-deficient Rabies can infect the motoneurons and express the fluorescent protein to label them but cannot spread. The second neurotropic virus in the cocktail is an Herpes Simplex Viral vector that encodes G (HSV-G). If co-injected with the G-deficient Rabies, both vectors will infect seed motoneurons and the production of G enabled by HSV-G will now complement the Rabies G-deficiency and allow the neoformation of viral particles in seed motoneurons and their spread through the synapse to presynaptic neurons, by definition premotor neurons. The beauty of the system is that the spread of the G-deficient Rabies vector will stop in premotor neurons for lack of G-complementation there as the G-complementing HSV cannot itself cross the synapse. In short, this method identifies strictly, for a given injected muscle, both its innervating motoneurons and their presynaptic partners (i.e. premotor neurons). In the

present study, it comes that virally labeled motoneurons are *bona fide* geniohyoid motoneurons and all other virally labeled neurons outside of motoneurons are *bona fide* geniohyoid premotor neurons.]

We have now reformulated the result para:

Line 280: “Alternatively, in keeping with both the obstructive nature of the apneas promoted by the 4A mutation and the presence of inhibitory candidate neurons in the medulla reticular formation we next looked for evidence that the *Mafa*^{4A} mutation may rather target neurons in the motor arm of respiratory control. We thus looked for *Mafa*⁺ inhibitory neurons in the circuit motorizing an upper airway opener muscle by trans-synaptic viral tracing⁴⁵. To do so, we injected unilaterally, in *Mafa*^{Lacz/+} pups at P3, a G-defective rabies virus variant encoding the fluorophore mCherry together with a helper virus encoding G and the fluorophore YFP in the geniohyoid muscle (i.e. that pulls the hyoid bone forward to increase airway patency) (**Fig.4e**) and examined their brains on P9. As expected, the only seed neurons (i.e. that co-express both fluorophores) were found in the accessory hypoglossal motor nucleus^{45,46} while the bulk of premotor neurons, presynaptic to the seed motoneurons (i.e. that express only the rabies virus encoded mCherry) was located in the dorsal part of the intermediate reticular formation in overlap with the *peri12* group of *Mafa*⁺ inhibitory neurons (**Fig.4f**)....”

Overall, this study contains a considerable amount of very interesting data for the field. However, the manuscript needs significant re-interpretation of the data, and additional experiments focused on the inclusion and analysis of the ECG and tongue EMG activity would provide much needed additional insights to the mechanistic and disease-related aspects of this mutation.

We hope that the provided additional experimental data supports the view that cellular targets of the mutation affect the motor rather than the sensory control arm of the respiratory behavior and that the expanded discussion of our data in relation to Rett syndrome (in response to reviewer#1) have provided additional insights to the mechanistic and disease-related aspect of the *Mafa*^{4A} mutation.

...f Joint plethysmographic (top trace) and click audio recordings (bottom trace) during an epoch of breath holding apneas showing that clicks that follow a post-inspiratory (arrowheads) breath holding apnea are associated to small upward ventilatory artefacts (downward arrows) that immediately precede onset of an expiration (e below the click trace, 4 of 5 clicks,) deflating the lung or occasionally an inspiration (i below the click trace, 1 of 5 clicks,) further inflating the lung.

Figure 2, Lecoin *et al.*

Figure 2. *Mafa*⁺ neurons of the reticular formation in the caudal medulla are inhibitory. **a** Whole mount (ventral view) of a *Mafa*^{LacZ/+} brain at E15 showing *Mafa* expression (Xgal staining, dark blue) restricted to the olfactory bulb (O.B.), to the pre-optic area of the hypothalamus (P.O.A.) to the brainstem (pons and medulla) and to the spinal cord. **b** hemi-transverse section (medial left, dorsal top) of a *Mafa*^{LacZ/+} pup at P0 showing that the preBötC area, below ChAT⁺ neurons (white) of the nucleus ambiguus (nAmb), is devoid of βgal⁺ cells (red). **c** same as b showing the absence of βgal⁺ cells (red) in the RTN, below ChAT⁺ neurons (white) of the facial motor nucleus (7Mo) - **d** transverse section of a *Mafa*^{LacZ/+} pup at P0 (at the d axial position indicated in a) showing *Mafa*⁺ cells immunostained for βgal (black) in the caudal medulla in the spinal part of the trigeminal nucleus (sp5), in the peri12 reticular formation laterally to the hypoglossal motor nucleus (Mo12), in motoneurons

of the accessory hypoglossal motor nucleus (acc-Mo12), **e** Close-up view of Mafa⁺ peri12 neurons (red) counterstained for ChAT⁺ (black) motoneurons at the same axial level as (d). **f** transverse section (at the f axial position indicated in a) showing Mafa⁺ cells in the Sp5 and medial to the nucleus ambiguus in the peri-nAmb area. **g** close up view of a hemi-transverse section (medial left, dorsal top) showing Mafa⁺ cells (red labeling) combined to Gad1 in situ hybridization (black) at the same axial level as (f) showing Gad1⁺/Mafa⁺ cells in the peri12 and peri-nAmb. **h** close-up view from the h inset in g. **i** close-up view from the i inset in g. **j** Quantification of neurochemical types of Mafa⁺ neurons of the caudal medulla reticular formation by in situ hybridization for vGlut2 (13 section counts), Glyt2 (11 section counts), Gad1 (10 section counts) and Gad2 (12 section counts). Scale bars (̳m): 100 (b-g), 20 (h,i). Note that Mafa⁺ neurons of the caudal reticular formation (peri12 and peri-nAmb regions) are almost exclusively inhibitory.

Fig.4, Lecoin *et al.*

Figure 4. Absence of *Mafa* inhibitory neurons in sensory and pontine respiratory structures and presence of *Mafa* inhibitory premotor neurons controlling the geniohyoid muscle. **a** left, transverse sections of P0 wild type pups showing the vagal jugular and nodose ganglia stained for DAPI (white) and by RNAscope using a probe against *Mafa* (red). Right, top, close-up views from the jugular ganglion probing *Mafa* (red), *vGUT2*(green), *Phox2b* (blue) and the merge. Right, bottom, close-up views from the nodose ganglion. **b** transverse sections of the trigeminal ganglion stained for DAPI (white) and *Mafa* (red). Right, close up views of the inset in **b** probing *Mafa* (red), *vGUT2* (green) and *Gad1&Gad2* (blue). Note the absence of *Mafa* expression in the nodose ganglion, expression of *Mafa* in the jugular

(sparse) and trigeminal ganglia restricted to *vGLUT2*-expressing excitatory neurons. **c** Summary of RNAscope probing for co-expression of *Mafa/vGLUT2*- and *Mafa/GAD1&2*-expression in vagal (5 sections, n=3 ganglia) and trigeminal (7 sections, n=3 ganglia) ganglia, nucleus tractus solitarius (NTS, 8 sections, n=2 pups) and peri12 (8 sections, n=2 pups). **d** left, transverse hemi-section of the brainstem at pontine level of a *VGAT^{Cre/+}; Mafa^{FloxLacZ/+}* P8 pup, showing absent *Mafa*⁺ inhibitory neurons (black) in the PB/KF, Pr5 areas but their presence in the SPO and LLM. Right, representative transverse hemi-section of the brainstem at medullary level showing the presence of *Mafa*⁺ inhibitory neurons in the peri12, peri-nAmb but their virtual absence in the Sp5, Pa5 and nTS (n=2 pups). **e** Monosynaptic tracing scheme showing unilateral injection of G-deleted Rabies viruses encoding mCherry and of a helper G- and YFP-encoding HSV virus into the geniohyoid muscle to transynaptically trace the position of premotor neurons. **f** Representative transverse section immunostained for β gal (white) in the caudal medulla showing mCherry⁺ virally labeled premotor neurons (red) and rare (at this axial level) seeding mCherry⁺/YFP⁺ geniohyoid motoneurons (yellow) in the accessory hypoglossal motor nucleus (arrowhead). **g** Close-up view of the inset in (f) showing that a fraction of mCherry⁺ premotor neurons indicated by arrows are *Mafa*⁺ (mCherry⁺/ β gal⁺, white nuclear labeling). **h** 2D-reconstruction of the position of *Mafa*⁺ geniohyoid premotor neurons (red dots) and *Mafa*⁺ cells (grey dots) showing their location in the peri12 area. Abbreviation: DLL, dorsal nucleus of the lateral lemniscus; KF, Kölliker-Fuse nucleus; PB, parabrachial nuclei; Pa5, paratrigeminal nucleus; Pr5, principal sensory trigeminal nucleus; scp, superior cerebellar peduncle; SPO, superior paraolivary nucleus. Calibrations (μ m): 100 (a,b left, c), 50 (a,b right), 1000 (d), 500 (f,h), 100 (inset).

Supplementary Fig.3, Lecoin *et al.*

Supplementary Figure 3. Clicks are time-locked to respiratory phase transitions during vocal breathing and breath holding apneas but not during eupnea nor central apneas. a-d Joint audio and plethysmographic analysis of the occurrence of clicks during the respiratory cycle in different breathing modes: vocal (a), breath-holding apneic (b), eupneic (c) or central-like apneic (d). a left panel, time distribution histogram of clicks (top, binning 20ms), occurrence of ultrasonic vocalizations (USV's, red, middle) and superimposed normalized plethysmographic (Pleth, bottom) black traces ($n=47$ from 4 wildtype P0 pups) synchronized (vertical red line) on the expiratory (exp) / inspiratory (insp., upward deflection) transition; right panel, same with synchronization on the peak compression of the lung that precedes USV's emission ($n=34$ for 4 wildtype P0 pups). Note that during vocal breathing virtually all clicks are time-locked to inspiratory (left panel) or expiratory (right panel) on-switch. b left panel, joint recordings of sound waveform signal (Sound) and plethysmography (Pleth) in two example *Mafa*^{n4A/n4A} mutant P0 pups (#1, #2) during a breath holding apnea. Note that clicks are time-locked to inspiratory on switch and to an upward artefact on the plethysmographic trace that precedes the first expiratory effort deflating the lung (right, $n=27$ superimposed breath holds from 5 *Mafa*^{n4A/n4A} P0 pups); right panel, time distribution histogram of clicks (top, binning 40ms) in register with synchronized and normalized plethysmographic artefacts (bottom). c Absence of clicks during eupneic breathing (40 superimposed respiratory cycles from 3 P0 pups). d Absence of clicks during central-like apneas. e summary histogram of the frequency of clicks in different breathing modes, the frequency of clicks is significantly, increased during vocal breathing (unpaired Student t-test $P=0.041$, $df=10$, $t=3.695$), breath holding apneas ($P=0.0033$, $df=10$, $t=3.834$), and not significantly changed during centra-like apneas ($P=0.083$, $df=1.16$, $t=1.921$) when compared to eupneic breathing.

Supplementary Figure 6. PreBötC neurons transiently express Mafa at embryonic stage and Peri12 Mafa neurons are V1 type interneurons. a Medulla transverse hemi-section of an intersectional *VGAT^{Cre/+};Mafa^{Flpo/+};RC::Fela/+* mutant pup at P0 showing the distributions of *Mafa⁺/VGAT⁺* inhibitory (beta-gal, red) and of *Mafa⁺/VGAT⁻* (GFP, green) neurons in the Peri12 and peri-nAmb region. b Recombination scheme in the *R26^{Fela}* allele. c Close-up view of the c inset in (a) counterstained for ChAT (blue) showing Peri12 *Mafa⁺* inhibitory neurons (red) and *Mafa⁺/ChAT⁺* motoneurons in the ventral Mo12 and its accessory nucleus (accMo12). d Close-up view of the d inset in (a) showing *Mafa⁺* neurons with a history of expression of *VGAT* (red) in the Peri-nAmb. and in the *preBötC*. e Similar close-up view as d in a *Mafa^{LacZ/Flpo};RC::Fela/+* mutant pup at P0 showing that *preBötC* neurons with a history of expression of *Mafa* (GFP) no longer express *Mafa* at P0 (absent red beta-gal labeling) while neurons in the Peri-nAmb maintain *Mafa* expression (arrows). f Hindbrain flatmount (anterior at top) at E10 showing *Mafa*-expressing territories as continuous anterior-posterior stripes of expression on either side of the midline. g Same at E12 showing that *Mafa* expression has down regulated except in most anterior and posterior aspect of the rhombencephalon. h Medulla hemi-section of an intersectional *VGAT^{Cre/+};Mafa^{Flpo/+};RC::hM3Dq/+* mutant pup showing the distribution of *hM3Dq*-mCherry expressing (red) and *Mafa⁺/VGAT⁻* (GFP, green) neurons. i Transverse section of the caudal medulla of a *En1^{Cre/+};RC::FL-tdT/+;Mafa^{LacZ/+}* pup at P0 showing *V1* type neurons (red), *Mafa⁺* cells (green) and *Mafa⁺/V1* type neurons (yellow). j Close-up view from the inset in (a) showing that more than half *Mafa⁺*

neurons (M) of the Peri12 area have a history of expression of En1 (ME) thus are V1 type neurons. k Quantification histogram showing that $67.8 \pm 1.7\%$ of all *Mafa*⁺ neurons in the Peri12 reticular formation (P.12) but only $10.7 \pm 4.5\%$ in the Peri-nAmb are V1 neurons (n= 3 pups). Scale bars (̳m): 250 (a,f), 100 (c,d,e,j), 200 (i).

Supplementary Table 4

Distribution of conditional *Mafa* mutants at birth (P0)

Crossing	Mafa ^{flox4A/+} x Mafa ^{flox4A/+}		
Genotype	Mafa ^{+/+}	Mafa ^{flox4A/+}	Mafa ^{flox4A/flox4}
Observed	18	34	23
Expected	18	37	18
Xhi ²	P=0.7013 Xhi ² =0.7096 df=2		
Survival	100%	93%	+100%

REVIEWER COMMENTS

Reviewer #1 (Remarks to the Author):

Previous review:

The authors show that the transcription factor Mafa is non-essential for survival and yet a specific mutation of this protein (Mafa4A) causes massive postnatal lethality. The authors attribute the animals' rapid demise to their inability to breathe normally and they identify the specific breathing anomaly elicited by this mutation as apneas of the breath-holding type possibly caused by airway obstruction. By restricting the Mafa4A mutation to neurons, the authors demonstrate that the lethality and the breathing disorder are of neural origin. They show that these apneas are not caused by a central chemoreceptor anomaly because the hypercapnic chemoreflex is preserved and the retrotrapezoid nucleus which is responsible for a large portion of this homeostatic reflex does not express Mafa. In addition, the authors show that the core of the breathing pattern generator (core circuit responsible for breathing automaticity) is minimally affected given that the inspiratory neural outflow observed in isolated brainstems maintained in vitro is normal. These observations lead the authors to the logical conclusion that the defect could be related to airway control (some form of obstructive apnea). Using a mutant created for this purpose (MafafloxLacZ26) they show that Mafa is heterogeneously expressed in the brainstem and is principally expressed by inhibitory (gad2+) neurons. They also provide evidence that Mafa4A mutation enhances gad2 expression. Evidence that the apneas result from excessive GABA release rests on two converging lines of evidence. First the incidence of apneas is reduced by acute administration of a subconvulsive dose of pentylenetetrazole, a noncompetitive antagonist of the gamma-aminobutyric acid (GABA)(A) receptor complex. Second, selective activation of Mafa-expressing inhibitory neurons (VGAT^{cre/+};Mafa^{Flpo/+} mice crossed with R26FL-hM3Dq/FL-hM3Dq to express a Gq-coupled DREADD exclusively in neurons positive for both VGAT and Mafa) reproduces the breathing disorder observed in Mafa4A/n4A mutants. Finally, the authors demonstrate, using a retrograde transsynaptic vector, that Mafa is expressed by a population of inhibitory premotor neurons that control the genio-hyoid muscle suggesting that these particular Mafa-expressing inhibitory neurons could be responsible for the postulated airway obstructions.

This work is outstanding, conceptually and technically. This team of investigators are top experts in brainstem development and breathing physiology. Their methodology (transgenic mice, vectors, techniques to record breathing in neonate mice, histology etc.) is state of the art. The discussion is excellent.

Re-review:

In revision the authors have provided additional evidence to suggest that the most prevalent apneas observed in the Maf4a mutants have an obstructive component. The absence of "exacerbated inspiratory or expiratory effort" (line 125) associated with these apneas suggest that these obstructive apneas are atypical, possibly that some simultaneous true breath holding phenomenon occurs simultaneously. However, the importance of the present study does not hinge on whether the type of apneas seen in these mutants is a strict phenocopy of the obstructive apneas associated with sleep apnea.

In my first review I suggested that the authors consider alternative interpretations for the apneas. Specifically, I raised the possibility that the Maf4a mutation could cause airway closure and or breath hold by altering the function of neurons located in the dorsolateral pons and within

regions such as the trigeminal nucleus or NTS that mediate the diving and sneezing reflexes or reflexes triggered from portion of the airways. After careful consideration the authors have rejected these hypotheses. The main argumentation is that the breathing defect is caused by an abnormality of Mafa-expressing inhibitory neurons which are absent from these brain regions. This reasoning is sound, especially when paired with the evidence that directly implicates the GABAergic neurons of the perihypoglossal region in the particular type of apnea present in the pups.

A couple of minor issues related to the statistics still need some attention.

- Please indicate in the methods which tests were used to ascertain that the data points were normally distributed. This is a requirement for the use of ANOVA and t-test.
- Figure 3b (effect of PTZ on apneas in the Mafa^{n4a/n4a}). I believe that a two-way ANOVA should be used to test whether PTZ is more effective than NaCl in the Mafa^{n4a/n4a} mutant strain (last four bars of the plot). If this first step shows overall statistical significance, then a posthoc test can be applied. Please verify with a statistician.

Finally:

- Line 796: perhaps Chi square or X² rather than X^{hi2}?
- Lines 90 and 91: Ser 64 or Ser 65 in both places?

Reviewer #2 (Remarks to the Author):

While there is a nice correlation between auditory broadband clicks and the “breath holding apneas”, the use of a click to demonstrate an obstructive apnea is not fully convincing. First, and foremost, the origin of the click sound in neonatal mice is unknown (as stated on page 18, 436), so how can they be used to demonstrate that a collapsed airway is re-opening? Additionally, their use raising other questions and missing quantification. Examples of this include:

- 1) In the right panel of Figure S3 the authors state that the click occurs at the transition from maximal expiration to USV onset (when the lung is “compressed” and then opens). Yet the change in pressure that is used to define expiration is a positive slope, which is opposite to how expiration is defined in the left panel of S3 and during the basal respiratory recordings in Fig 1. So in this case, why is the click not occurring at the onset of inspiration? Why are the USVs occurring during inspiration? This confusion is especially important since the click preceding the deflation is used as evidence of airway opening during a collapsed expiratory state.
- 2) Following this point, why is it that the pressure change is called an “artefact”? If it isn’t an artefact, then it would be that the click is associated with a small inspiration. Perhaps this is needed to “open” the airway to enable the completion of expiration? Nonetheless, this discussion shows that while a click may be a feature of a breath hold, it is difficult to interpret that it demonstrates airway collapse.
- 3) The percent of breath holds with and without clicks is never described. If the clicks happen in just a few breath holds, then they should be used as a defining feature.

All together, the manuscript uses state of the art genetic and molecular techniques to identify an inhibitory premotor population that might control upper airway patency and explain a “breath holding” apnea phenotype. These results are meaningful and should be published. The data in

the paper is complete. However, the claim that the phenotype observed is an obstructive apnea is still a model. If this cannot be experimentally addressed, the authors should refrain from using “obstructive apnea” in the title and as definitively shown throughout the results. Instead, perhaps this model and the use of clicks in perhaps showing this, should be a topic in the discussion.

Reviewer #3 (Remarks to the Author):

I have no additional comments on this revised MS. The authors have sufficiently addressed all of my concerns.

Response to reviewers

[Referee comments are included in italics, our responses are in blue and text changes in red]

REVIEWER COMMENTS

We would like to thank the reviewers for their positive assessment of our work and their constructive comments.

Reviewer #1 (Remarks to the Author):

Previous review:

The authors show that the transcription factor Mafa is non-essential for survival and yet a specific mutation of this protein (Mafa4A) causes massive postnatal lethality. The authors attribute the animals' rapid demise to their inability to breathe normally and they identify the specific breathing anomaly elicited by this mutation as apneas of the breath-holding type possibly caused by airway obstruction. By restricting the Mafa4A mutation to neurons, the authors demonstrate that the lethality and the breathing disorder are of neural origin. They show that these apneas are not caused by a central chemoreceptor anomaly because the hypercapnic chemoreflex is preserved and the retrotrapezoid nucleus which is responsible for a large portion of this homeostatic reflex does not express Mafa. In addition, the authors show that the core of the breathing pattern generator (core circuit responsible for breathing automaticity) is minimally affected given that the inspiratory neural outflow observed in isolated brainstems maintained in vitro is normal. These observations lead the authors to the logical conclusion that the defect could be related to airway control (some form of obstructive apnea). Using a mutant created for this purpose (MafafloxLacZ26) they show that Mafa is heterogeneously expressed in the brainstem and is principally expressed by inhibitory (gad2+) neurons. They also provide evidence that Mafa4A mutation enhances gad2 expression. Evidence that the apneas result from excessive GABA release rests on two converging lines of evidence. First the incidence of apneas is reduced by acute administration of a subconvulsive dose of pentylenetetrazole, a noncompetitive antagonist of the gamma-aminobutyric acid (GABA)(A) receptor complex. Second, selective activation of Mafa-expressing inhibitory neurons (VGATcre/+;MafaFlpo/+ mice crossed with R26FL-hM3Dq/FL-hM3Dq to express a Gq-coupled DREADD exclusively in neurons positive for both VGAT and Mafa) reproduces the breathing disorder observed in Mafan4A/n4A mutants.

Finally, the authors demonstrate, using a retrograde transsynaptic vector, that Mafa is expressed by a population of inhibitory premotor neurons that control the genio-hyoid muscle suggesting that these particular Mafa-expressing inhibitory neurons could be responsible for the postulated airway obstructions.

This work is outstanding, conceptually and technically. This team of investigators are top experts in brainstem development and breathing physiology. Their methodology (transgenic mice, vectors, techniques to record breathing in neonate mice, histology etc.) is state of the art. The discussion is excellent.

Re-review:

In revision the authors have provided additional evidence to suggest that the most prevalent apneas observed in the Maf4a mutants have an obstructive component. The absence of "exacerbated inspiratory or expiratory effort" (line 125) associated with these apneas suggest that these obstructive apneas are atypical, possibly that some simultaneous true breath holding phenomenon occurs simultaneously. However, the importance of the present study does not hinge on whether the type of apneas seen in these mutants is a strict phenocopy of the obstructive apneas associated with sleep apnea.

In my first review I suggested that the authors consider alternative interpretations for the apneas. Specifically, I raised the possibility that the Maf4a mutation could cause airway closure and or breath

hold by altering the function of neurons located in the dorsolateral pons and within regions such as the trigeminal nucleus or NTS that mediate the diving and sneezing reflexes or reflexes triggered from portion of the airways. After careful consideration the authors have rejected these hypotheses. The main argumentation is that the breathing defect is caused by an abnormality of *Mafa*-expressing inhibitory neurons which are absent from these brain regions. This reasoning is sound, especially when paired with the evidence that directly implicates the GABAergic neurons of the perihypoglossal region in the particular type of apnea present in the pups.

We are grateful for these positive comments on our work.

A couple of minor issues related to the statistics still need some attention.

- Please indicate in the methods which tests were used to ascertain that the data points were normally distributed. This is a requirement for the use of ANOVA and t-test.

We kindly thank the reviewer for pointing this omission. We have now indicated that D'Agostino & Pearson and Shapiro-Wilk tests were used to test normality.

Text change:

In Materials and Methods: Statistical Analysis

Line 782... "Normal distributions of data points were tested using D'Agostino & Pearson or Shapiro-Wilk tests."

- Figure 3b (effect of PTZ on apneas in the *Mafa*^{n4A/n4A}). I believe that a two-way ANOVA should be used to test whether PTZ is more effective than NaCl in the *Mafa*^{n4A/n4A} mutant strain (last four bars of the plot). If this first step shows overall statistical significance, then a posthoc test can be applied. Please verify with a statistician.

Thank you, our experimental design was a repeated measures design in order to maximize statistical power and minimize the number of animals used in accordance with local and European guidelines for the use of animals in scientific experimentation. The repeated measures design allowed each mouse to be its own control minimizing one source of experimental variability. In addition, the effect of PTZ was tested at P0+12 hours when the apneic time fraction is maximal, thereafter pups begin to die which precluded a multi-treatment approach. We have consulted a biostatistician in our department with regards to the appropriate statistical analyses.

For the results in panel 3b, in a first experiment, *Mafa*^{n4A/n4A} mutant mice were evaluated before and after PTZ. In a second independent experiment, we verified using the same method whether the vehicle had any effect on apneic time fraction in *Mafa*^{n4A/n4A} mutant mice. All four groups of data passed normality test before (P=0.3869) and after (P=0.1758) PTZ as well as before (P=0.7249) and after (P=0.4956) NaCl. Since sphericity and normality criteria were fulfilled, a repeated measures two-way ANOVA has now been performed, showing that treatment has a significant effect $F(1,18)=8.176$ $P=0.0104$, and that both groups of *Mafa*^{n4A/n4A} mutant mice were comparable $F(1,18)=1.139$, $P=0.2999$. No interference was detected $F(1, 18)=1.452$, $P=0.2438$. Then, Bonferroni's multiple comparisons showed that the mean (untreated-treated) difference of 14.63 observed in PTZ treatment was significant ($P=0.0202$) whereas the 5.954 difference observed in NaCl treatment was not ($P=0.5147$).

Thanks to your question we have systematically verified all statistics and plots and corrected in panel 3b the erroneous (**) for (*) for PTZ effect on apneas in *Mafa*^{n4A/n4A} mutant pups that previously should have corresponded to $p=0.0363$ of the then performed Student's t-test and that now corresponds to $p=0.0202$ of the Bonferroni test. We corrected a slip in panel 3c for the effect of PTZ on breath holding apneas of control pups, (**) replace (ns) to match the $p=0.0039$ value of Wilcoxon test (now mentioned in the legend and replacing the previous Student's t-test) after verifying that the dataset was not normally distributed.

Text change:

Line 1131...Legend of Figure 3: “b PTZ has no effect on the apneic time fraction of control pups (grey symbols, Wilcoxon paired test, P=0.921) but PTZ and not NaCl reduces the apneic time fraction of *Mafa*^{n4A/n4A} mutant pups (red symbols, two-way ANOVA followed by Bonferroni’s multiple comparison test, P=0.0202 for PTZ; P=0.5147 for NaCl).”

Finally:

- Line 796: perhaps Chi square or X2 rather than Xhi2?

Thanks, we have replaced throughout (Material and Methods: Statistical analysis; Supplementary Tables 1,3 and 4) Xhi2 by χ^2 .

First occurrence:

Line 785...“Chi square (χ^2) test was used for discrete variables.”

- Lines 90 and 91: Ser 64 or Ser 65 in both places?

Thank you, there is hidden subtlety in our statement. The human mutation S64F (Ser64Phe) prevents Mafa phosphorylation on Ser65 by a “priming kinase” and all subsequent phosphorylations of Mafa by GSK3 kinase on 4ser/thr residues (S61, T57, T53, S49).

Text change

Line 88... “More precisely, phosphorylation is a key regulatory mechanism, as shown by the Ser64Phe human mutation that impairs Mafa phosphorylation while causing familial diabetes mellitus and insulinomatosis²². This mutation prevents Mafa phosphorylation on Ser65 by a priming kinase and all subsequent phosphorylations of Mafa by GSK3 kinase on 4 Ser/Thr residues. These 4 residues (S61, T57, T53, S49) are highly conserved among MAF family members, and their phosphorylation mediates degradation of the protein through the proteasome^{21,23}.”

Reviewer #2 (Remarks to the Author):

While there is a nice correlation between auditory broadband clicks and the “breath holding apneas”, the use of a click to demonstrate an obstructive apnea is not fully convincing.

We agree with this reviewer that the demonstration that breath holding apneas have an obstructive nature is not provided and that “obstructive” shouldn’t be used to qualify breath holding apneas. As suggested we keep it a matter for discussion, all the more justified after responding to your other queries (see detail in response to your last point).

First, and foremost, the origin of the click sound in neonatal mice is unknown (as stated on page 18, 436), so how can they be used to demonstrate that a collapsed airway is re-opening?

We agree that mechanisms generating clicks in the airway have not been studied in juvenile mice. The idea that click elicitation requires a preliminary enclosure of air in a compartment comes from modeling efforts to determine critical parameters of clicks in Dolphins and from phonological studies of human languages. We agree with the reviewer that clicks have no demonstrative value (see below).

Text changes:

Line 181... ...we focused on brief (ms order) broad-band audio-mechanical events named “clicks” or cracking sounds (see refs ^{9,30,31}) that critically rely on openings of airway cavities from a closed state³².”

Line 436... ... and the final elicitation of a click, otherwise known to correspond to the opening of airway cavities from a closed state, a concept that has phonological relevance, in association to USV

emission in mice (this study and⁹) and as a regular part of the consonant systems of many human languages⁶⁰.”

Additionally, their use raising other questions and missing quantification. Examples of this include: 1) In the right panel of Figure S3 the authors state that the click occurs at the transition from maximal expiration to USV onset (when the lung is “compressed” and then opens). Yet the change in pressure that is used to define expiration is a positive slope, which is opposite to how expiration is defined in the left panel of S3 and during the basal respiratory recordings in Fig 1.

We thank the reviewer for raising this concern, we would like to briefly explain the reasons why we would like to maintain our statement (and the reason why we have explicitly named the successive phase in the figure S3). Since the modeling of Drorbaug & Fenn, 1955, the plethysmographic signal is known to record not only volumes (from the vaporization of water entering the lungs, ideal gas law) but in addition compression ($P_{\text{subglottal}} - P_{\text{plethysmo}}$ proportional to airway resistance (Raw) multiplied the air flow in the airway, Boyle law).

It comes that measurement of V_T (the goal of the plethysmography) can only be rigorously done at times of zero flow which generally happens twice during the respiratory cycle: at the end of inspiration (flow inversion) and at the end of expiration (zeroing to baseline when passive or inversion of flow when active). Hence, the measuring of V_T between these two points. Note that a third period where V_T may be measured is throughout breath holds as the flow is likely close to zero. During vocal breathing, the plethysmograph mostly detects the dramatic increase of subglottal pressure during compression (negative slope) followed after the click by a decrease of this pressure (positive slope) due to opening of the airway leading to air outflow powering USV emission. In contrast, recovery from the inspiratory vaporization of water (negative slope) plays only a minor role there (if any). In other words, the time course of the plethysmographic signal indicates that, at end compression, the pressure due to vaporization is negligible compared to that due to compression. We do not consider it necessary to add these detailed considerations to the ms, we simply have added one sentence in the methods highlighting the role of lung compression with a reference to Drorbaug and Fenn, 1955, our Ref. 92).

Text change

Line 628... “Note that the analysis of plethysmographic recordings during vocalizations (Supplementary Figure 3A), must prominently take into account lung compression, thus the relationship between pressure, flow rate and resistance of upper airways⁹² (Boyle law) in addition to the vaporization of water during inspiration (Ideal gas law) itself prominent during resting breathing.”

So in this case, why is the click not occurring at the onset of inspiration?

From the preceding comment we do not follow the logic of the question.

This said, it is true that clicks are also elicited at the beginning of inspiration in both behavioral contexts of vocalization as shown on (panel a, left) or breath-holding (panel b, left) of Supplementary Figure 3. This was presented and commented in Mat & Meth (lines 632 – 636).

Why are the USVs occurring during inspiration? This confusion is especially important since the click preceding the deflation is used as evidence of airway opening during a collapsed expiratory state.

We have explained above why our interpretation is that clicks are timed to max lung compression and not to inspiration.

2) Following this point, why is it that the pressure change is called an “artefact”? If it isn’t an artefact, then it would be that the click is associated with a small inspiration. Perhaps this is needed to “open” the airway to enable the completion of expiration?

We thank the reviewer who has drawn our attention to this point.

Both inspirations (in pups) and clicks (in dolphins and humans) are known to involve generation of a negative pressure within a compartment of the airways, so that both should generate an upward

deflection of the plethysmographic trace. As you rightly picked up, because the origin of clicks is unknown we have now opted for a purely descriptive formulation and replaced “artefact” by “upward pressure shift” throughout.

Double microphonic-plethysmographic recordings show that small amplitude upward pressure shifts occur simultaneously to a click as if both were initiated by the same brief event in the respiratory track. It remains that click-associated upward pressure shifts may either correspond to “small inspirations” as proposed or to direct detection by the plethysmograph of the pressure wave underlying the click sound. Support for the latter comes from the shape of the upward shift that is distinct from that of the smallest inspirations in both size and time constant (provided that filtering inherent to the plethysmographic method is not too dramatic), please see a clearer illustration below. An argument against the former comes from yet unobserved (to our knowledge) scaled inspiratory/post-inspiratory motor drives necessary to respectively produce the transient inflow (“small inspiration”) and prevent lung collapsing during the immediately following expiratory outflow terminating breath holds.

The previous use of the misnomer “artefact” as a shorthand in legends probably reflected our inclination, an Ockham’s razor of sort, towards the non-respiratory hypothesis (we mean here a pressure shift with respiratory timing but without inspiratory role per se).

Text change

Line 188... ...systematically time locked to small amplitude upward pressure shifts of the plethysmographic trace (downward arrows in Figs 1f, 3e, 6b and Supplementary Fig.3b) that immediately precede expiratory-like lung deflations terminating breath holds.”

See also corrections in legends of Figures 1, 3, 6 and Supplementary Figure 3).

Nonetheless, this discussion shows that while a click may be a feature of a breath hold, it is difficult to interpret that is demonstrates airway collapse.

We hope that the elements provided in response to your point 1 make our interpretation more compatible with an airway collapse model, again we agree that clicks have no demonstrative value.

Text change in the discussion:

Line 429... Although clicks have unknown origin, in these two breathing contexts they could strikingly stand as acoustic signatures of the re-opening of closed airways.”

3) The percent of breath holds with and without clicks is never described. If the clicks happen in just a few breath holds, then they should be used as a defining feature.

We surmise the reviewer meant “should not”.

Thank you, the fraction of breath holding apneas associated to a click is about 70%. In face of the virtual absence of clicks during eupnea and central-like apneas, clicks can reasonably be considered a robust (if not a defining) feature of breath holding apneas.

Text change:

Line 186... "We found that **like the vocalizations**, breath holding apneas were accompanied by clicks (**69.5% fraction of breath holds with click, n=46 breath holds, from 9 wildtype pups, Fig.1f**)...."

All together, the manuscript uses state of the art genetic and molecular techniques to identify an inhibitory premotor population that might control upper airway patency and explain a "breath holding" apnea phenotype. These results are meaningful and should be published. The data in the paper is complete. However, the claim that the phenotype observed is an obstructive apnea is still a model. If this cannot be experimentally addressed, the authors should refrain from using "obstructive apnea" in the title and as definitively shown throughout the results. Instead, perhaps this model and the use of clicks in perhaps showing this, should be a topic in the discussion.

We thank the reviewer for his comments and agree with his recommendations, we have changed the manuscript accordingly as follows :

We no longer qualify apneas as "obstructive" in the title, summary, introduction and results sections, except to make the reader understand why we studied breath holds or to conclude a result paragraph presenting arguments supporting the hypothetical obstructive nature of breath holding apneas. We also agree to keep the obstructive status of breath holding hypothetical in the discussion.

Text changes

The title:

Mafa-dependent GABAergic activity promotes neonatal apneas

In the summary:

We have withdrawn the first sentence and re-worded as follows:

Line 34... "We report that mice with a mutated form of the transcription factor Mafa **rapidly die after birth and** present with an abnormally high incidence of apneas including breath holding apneas **where a flattening of respiratory air flow delays completion of the expiration.**"

In the Results:

Line 193... "Thus, although a direct measurement of airway patency was not feasible in our **experimental conditions**, these data strongly support the obstructive nature of breath holding apneas in mutant pups."

In the discussion :

Line 430... "The present observations suggest that in mutants, stabilization of Mafa in inhibitory premotor neurons controlling upper airway patency might favor obstructions. We show that this may apply to the genioid muscle known to increase airway patency. In addition, we identify breath holds as major targets of Mafa-related controls. Although the obstructive nature of breath holds would require further investigation, it is strongly supported by the ability to maintain lung inflation, the resuming of rhythmic inflation efforts and the final elicitation of a click, otherwise known to correspond to the opening of airway cavities from a closed state, a concept that has phonological relevance, in association to USV emission in mice (this study and⁹) and as a regular part of the consonant systems of many human languages⁶⁰."

Line 447... "In *Mafa*^{n^{4A}/n^{4A}} mutant pups, the onset of the obstructive event **would take place** after inspiration has ceased in agreement with an impairment of the post-inspiratory coordination of pharyngeal muscles contractions albeit in the absence of noticeable joint expiratory efforts."

Line 485...“Thus, **if obstructive**, breath holding apneas of Mafa mutants likely originate in the pharynx mostly composed of soft tissues, thus most susceptible to collapse⁷⁶.”

All together, we hope to have (i) clarified better the respiratory timing of clicks and by doing so increased their suggestive (admitted not demonstrative) value of airway obstruction, and (ii) kept the obstructive nature of breath holding apneas our preferred model, in a manner most compatible with the statement by reviewer 1: “... authors have provided additional evidence to suggest that the most prevalent apneas observed in the Mafa4a mutants have an obstructive component.”

Reviewer #3 (Remarks to the Author):

I have no additional comments on this revised MS. The authors have sufficiently addressed all of my concerns.

We thank the reviewer for his assessment.